# LEARNING TO DESIGN RNA

**Frederic Runge**[1]*, **Danny Stoll**[1]*, **Stefan Falkner**[1,2] **& Frank Hutter**[1]
[1]Department of Computer Science, University of Freiburg
[2]Bosch Center for Artificial Intelligence, Robert Bosch GmbH
`{runget,stolld,sfalkner,fh}@cs.uni-freiburg.de`

## ABSTRACT

Designing RNA molecules has garnered recent interest in medicine, synthetic biology, biotechnology and bioinformatics since many functional RNA molecules were shown to be involved in regulatory processes for transcription, epigenetics and translation. Since an RNA's function depends on its structural properties, the *RNA Design* problem is to find an RNA sequence which satisfies given structural constraints. Here, we propose a new algorithm for the *RNA Design* problem, dubbed *LEARNA*. *LEARNA* uses deep reinforcement learning to train a policy network to sequentially design an entire RNA sequence given a specified target structure. By meta-learning across 65 000 different *RNA Design* tasks for one hour on 20 CPU cores, our extension *Meta-LEARNA* constructs an *RNA Design* policy that can be applied out of the box to solve novel *RNA Design* tasks. Methodologically, for what we believe to be the first time, we jointly optimize over a rich space of architectures for the policy network, the hyperparameters of the training procedure and the formulation of the decision process. Comprehensive empirical results on two widely-used *RNA Design* benchmarks, as well as a third one that we introduce, show that our approach achieves new state-of-the-art performance on the former while also being orders of magnitudes faster in reaching the previous state-of-the-art performance. In an ablation study, we analyze the importance of our method's different components.

## 1 INTRODUCTION

RNA is one of the major classes of information-carrying biopolymers in the cells of living organisms. Recent studies revealed a key role of functional non-protein-coding RNAs (ncRNAs) in regulatory processes and transcription control, which have also been connected to certain diseases like *Parkinson's disease* and *Alzheimer's disease* (ENCODE Project Consortium and others, 2004; Gstir et al., 2014; Kaushik et al., 2018). Functional ncRNAs are involved in the modulation of epigenetic marks, altering of messenger RNA (mRNA) stability, mRNA translation, alternative splicing, signal transduction and scaffolding of large macromolecular complexes (Vandivier et al., 2016). Therefore, engineering of ncRNA molecules is of growing importance with applications ranging from biotechnology and medicine to synthetic biology (Delebecque et al., 2011; 2012; Guo et al., 2010; Meyer et al., 2015). In fact, successful attempts to create functional RNA sequences *in vitro* and *in vivo* have been reported (Dotu et al., 2014; Wachsmuth et al., 2013).

At its most basic structural form, RNA is a sequence of the four nucleotides *Adenine (A)*, *Guanine (G)*, *Cytosine (C)* and *Uracile (U)*. This nucleotide sequence is called the *RNA sequence*, or primary structure. While the RNA sequence serves as the blueprint, the functional structure of the RNA molecule is determined by the folding translating the RNA sequence into its 3D tertiary structure. The intrinsic thermodynamic properties of the sequence dictate the resulting fold. The hydrogen bonds formed between two corresponding nucleotides constitute one of the driving forces in the thermodynamic model and influence the tertiary structure heavily. The structure that encompasses these hydrogen bonds is commonly referred to as the secondary structure of RNA. Many algorithms for RNA tertiary structure design directly work on RNA secondary structures (Kerpedjiev et al., 2015; Zhao et al., 2012; Reinharz et al., 2012). Therefore, fast and accurate algorithms for RNA secondary structure design could advance the current state of the art in RNA engineering.

---

*Frederic Runge and Danny Stoll contributed equally to this work; order determined by coinflip.

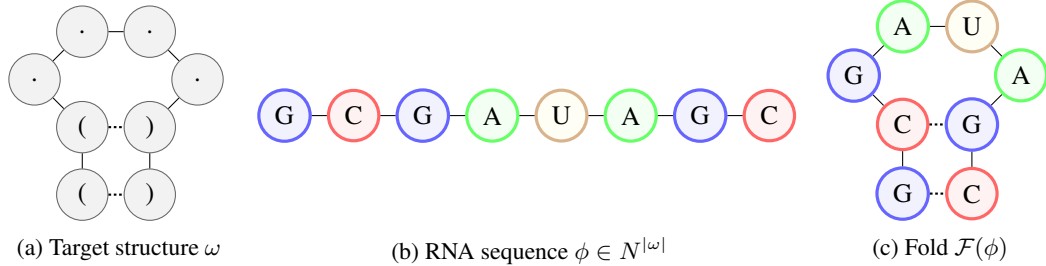

(a) Target structure $\omega$      (b) RNA sequence $\phi \in N^{|\omega|}$      (c) Fold $\mathcal{F}(\phi)$

Figure 1: Illustration of the *RNA Design* problem using a folding algorithm $\mathcal{F}$ and the dot-bracket notation. Given the desired RNA secondary structure represented in the dot-bracket notation (a), the task is to design an RNA sequence (b) that folds into the desired secondary structure (c).

The problem of finding an RNA sequence that folds into a desired secondary structure is known as the *RNA Design* problem or *RNA inverse folding* (Hofacker et al., 1994). Most algorithms for *RNA Design* focus on search strategies that start with an initial nucleotide sequence and modify it to find a solution for the given secondary structure (Hofacker et al., 1994; Andronescu et al., 2004; Taneda, 2011; Esmaili-Taheri et al., 2014; Eastman et al., 2018). In contrast, in this paper we describe a novel generative deep reinforcement learning (RL) approach to this problem. Our contributions are as follows:

- We describe *LEARNA*, a deep RL algorithm for *RNA Design*. *LEARNA* trains a policy network that, given a target secondary structure, can be rolled out to sequentially predict the entire RNA sequence. After generating an RNA sequence, our approach folds this sequence, locally adapts it, and uses the distance of the resulting structure to the target structure as an error signal for the RL agent.

- We describe *Meta-LEARNA*, a version of *LEARNA* that learns a single policy across many *RNA Design* tasks directly applicable to new *RNA Design* tasks. Specifically, it learns a conditional generative model from which we can sample candidate RNA sequences for a given RNA target structure, solving many problems with the first sample.

- Since validation in *RNA Design* literature is often done using undisclosed data sources (Eastman et al., 2018; Yang et al., 2017) and previous benchmarks do not have a training split associated with them (Taneda, 2011; Anderson-Lee et al., 2016; Kleinkauf et al., 2015), we introduce a new benchmark dataset with an explicit training, validation and test split.

- We jointly optimize the architecture of the policy network together with training hyperparameters and the state representation. By assessing the importance of these choices, we show that this is essential to achieve best results. To the best of our knowledge, this is the first application of architecture search (AS) to RL, the first application of AS to meta-learning, and the first time AS is used to choose the best combination of convolutional and recurrent layers.

- A comprehensive empirical analysis shows that our approach achieves new state-of-the-art performance on the two most commonly used *RNA Design* benchmark datasets: Rfam-Taneda (following Taneda (2011)) and Eterna100 (following Anderson-Lee et al. (2016)). Furthermore, *Meta-LEARNA* achieves the results of the previous state-of-the-art approaches $63\times$ and $1765\times$ faster, respectively.

## 2 THE RNA DESIGN PROBLEM

RNA folding algorithms $\mathcal{F}$ map from an RNA sequence to a representation of its secondary structure. The *RNA Design* problem aims to find an inverse mapping for a given RNA folding algorithm $\mathcal{F}$:

**Definition 1** (RNA Design). *Given a folding algorithm $\mathcal{F}$ and a target RNA secondary structure $\omega$, the RNA Design problem is to find an RNA sequence $\phi \in N^{|\omega|} = \{A, G, C, U\}^{|\omega|}$ that satisfies $\omega = \mathcal{F}(\phi)$.*

In this paper, we employ the most common folding algorithm: the *Zuker algorithm* (Zuker & Stiegler, 1981; Zuker & Sankoff, 1984), which uses a thermodynamic model to minimize the free energy to find the most stable conformation of the RNA secondary structure. We note, however, that our approach is not limited to it and would also directly apply for any other RNA folding algorithm.

RNA secondary structures are often represented using the dot-bracket notation, where dots stand for unbound sites and nucleotides connected by a hydrogen bond are marked by opening and closing brackets.[1] Figure 1 illustrates the *RNA Design* problem and the dot-bracket notation.

Most algorithms for *RNA Design* employ a structural loss function $L_\omega(\mathcal{F}(\phi))$ to quantify the difference between the target structure $\omega$ and the structure resulting from folding an RNA sequence $\phi$. A minimizer of this loss corresponds to a solution to the *RNA Design* problem for a specified target structure $\omega$:

$$\phi^* \in \arg\min_{\phi \in N^{|\omega|}} L_\omega(\mathcal{F}(\phi)) \qquad . \tag{1}$$

A common loss function, which we also employ in this work, is the *Hamming distance* (Hamming, 1950) between two structures. We note that multiple RNA sequences may fold to the same secondary structure, such that the *RNA Design* problem does not generally have a unique solution; one could distinguish between solutions by preferring more stable folds, targeting a specific GC content, or satisfying other constraints; all of these could be incorporated into the loss function being optimized.

## 3 LEARNING TO DESIGN RNA

In this section we describe our novel generative approach for the *RNA Design* problem based on reinforcement learning. We first formulate *RNA Design* as a decision process and then propose several strategies to yield agents that learn to design RNA end-to-end.

### 3.1 MODELLING RNA DESIGN AS A DECISION PROCESS

We propose to model the *RNA Design* problem with respect to a given target structure $\omega$ as the undiscounted decision process $D_\omega := (\mathcal{S}, \mathcal{A}, \mathcal{R}_\omega, \mathcal{P}_\omega)$; its components (the state space $\mathcal{S}$, the action space $\mathcal{A}$, the reward function $\mathcal{R}_\omega$ and the transition function $\mathcal{P}_\omega$) are specified in the paragraphs below. The *RNA Design* problem is defined with respect to a folding algorithm, which we denote as $\mathcal{F}(\cdot)$; further, we denote the set of dot-bracket encoded RNA secondary structures with $\Omega$.

**Action space** In each episode, the agent has the task to design an RNA sequence that folds into the given $\omega \in \Omega$. To design a candidate solution $\phi \in N^{|\omega|}$, the agent places nucleotides by choosing an action $a^t$ at each time step $t$. For unpaired sites, $a^t$ corresponds to one of the four RNA nucleotides (G, C, A or U); for paired sites, two nucleotides are placed simultaneously. In our formulation, these two nucleotides correspond to one of the *Watson-Crick base pairs* (GC, CG, AU, or UA). At time step $t$, the action space can then be defined as

$$\mathcal{A} := \{0, 1, 2, 3\} \equiv \begin{cases} \{A, G, C, U\} & \text{for } \mathcal{C}_\omega(t) = . \quad [\text{"dot"}] \\ \{GC, CG, AU, UA\} & \text{for } \mathcal{C}_\omega(t) = ( \quad [\text{"opening bracket"}] \end{cases} , \tag{2}$$

where $\mathcal{C}_\omega(t)$ is the t-th character of the target structure $\omega$. There is no action for closing brackets, as the associated sites are assigned nucleotides when encountering the corresponding opening bracket. See Figure 2 for an illustration of the action rollout.

**State space** The agent chooses an action $a^t$ based on the state $s^t$ provided by the environment. We formulated states to provide local information to the agent. For this we set $s^t$ to the $(2\kappa + 1)$-gram centered around the t-th site of the target structure $\omega$, where $\kappa$ is a hyperparameter we dub the *state radius*. To be able to construct this centered n-gram at all sites, we introduced $\kappa$ padding characters at the start and the end of the target structure. Formally, the state space can then be written as

$$\mathcal{S} := \{0, 1, 2, 3\}^{2\kappa+1} \equiv (\mathcal{B} \cup \{\text{padding}\})^{2\kappa+1} , \tag{3}$$

where $\mathcal{B}$ is the set of symbols in the dot-bracket notation: a dot, an opening and a closing bracket.

---

[1]There are also other notations (Shapiro, 1988; Fontana et al., 1993); our approach would also apply to these.

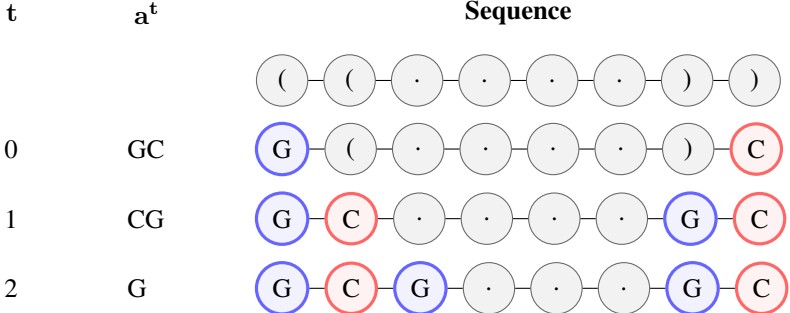

Figure 2: Illustration of an action rollout in the proposed decision process. The agent sequentially builds a candidate solution by choosing actions to place nucleotides. At paired sites, as indicated by a pair of brackets, two nucleotides are placed simultaneously ($t = 0$ and $t = 1$); while at unpaired sites a single nucleotide is placed ($t = 2$).

**Transition Function**   Since at each time step $t$ the state $s^t$ is set to a fixed $(2\kappa + 1)$-gram, the transition function $\mathcal{P}_\omega$ is deterministic and defined accordingly.

**Reward Function**   At the terminal time step $T$ the agent has assigned nucleotides to all sites of the candidate solution $\phi$ and the environment generates the (only non-zero) reward $\mathcal{R}_\omega^T(\phi)$. This reward is based on the *Hamming distance* $d_H(\mathcal{F}(\phi), \omega)$ between the folded candidate solution $\mathcal{F}(\phi)$ and the target structure $\omega$. We normalize this distance with respect to the sequence length $|\omega|$ to formulate the loss function $L_\omega(\mathcal{F}(\phi)) := d_H(\mathcal{F}(\phi), \omega) \, / \, |\omega|$. To solve the optimization problem in Equation 1, we set

$$\mathcal{R}_\omega^T(\phi) := (1 - L_\omega(\mathcal{F}(\phi)))^\alpha \qquad , \tag{4}$$

where $\alpha > 1$ is a hyperparameter to shape the reward. Additionally, we include a local improvement step to increase sample efficiency and boost performance of the stochastic RL agent as follows: If $d_H(\mathcal{F}(\phi), \omega) < \xi$, where $\xi$ is a hyperparameter, we search through neighboring primary sequences by exhaustively trying all combinations for the mismatched sites, returning the minimum *Hamming distance* observed. In our experiments, we set $\xi = 5$, which corresponds to at most $4^4 = 256$ neighboring sequences. Pseudocode for computing $\mathcal{R}_\omega^T(\phi)$ can be found in Appendix A.

### 3.2   Obtaining Policies for RNA Design

We use deep reinforcement learning to learn the parameters $\theta$ of policy networks $\pi^\theta$. Our policy networks consist of an embedding layer for the input state and a deep neural network; this neural network optionally contains convolutional, recurrent and fully-connected layers, and its precise architecture is jointly optimized together with the hyperparameters as described in Section 4. We propose several strategies to learn the parameters $\theta$ of a given policy network as detailed below.

**LEARNA**   The *LEARNA* strategy learns to design a sequence for the target structure $\omega$ in an online fashion, from scratch. The parameters $\theta$ are randomly initialized before the agent episodically interacts with the decision process $\mathcal{D}_\omega$. For updating the parameters we use the policy gradient method PPO (Schulman et al., 2017), which was recently successfully applied to several other problems (Heess et al., 2017; Bansal et al., 2018; Zoph et al., 2018).

**Meta-LEARNA**   *Meta-LEARNA* uses a meta-learning approach (Lemke et al., 2015) that views the *RNA Design* problems associated with the target structures in the training set $\Omega_{\text{train}}$ as tasks and learns to transfer knowledge across them. Each of the target structures $\omega_i \in \Omega_{\text{train}}$ defines a different decision process $\mathcal{D}_{\omega_i}$; using asynchronous parallel PPO updates, we train a single policy network on all of these. Once the training is finished, the parameters $\theta$ are fixed and $\pi^\theta$ can be applied to the decision process $\mathcal{D}_\omega$ by sampling from the learned generative model.

**Meta-LEARNA-Adapt** *Meta-LEARNA-Adapt* combines the previous two strategies: First, we obtain an initialization for the parameters $\theta$ by running *Meta-LEARNA* on $\Omega_{\text{train}}$. Then, when applied to the decision process $\mathcal{D}_\omega$, the parameters $\theta$ are further adapted using the *LEARNA* strategy.

# 4 JOINT ARCHITECTURE AND HYPERPARAMETER SEARCH

One problem of current deep reinforcement learning methods is that their performance can be very sensitive to choices regarding the architecture of the policy network, the training hyperparameters, and the formulation of the problem as a decision process (Henderson et al., 2017). Therefore, we propose to use techniques from the field of *automatic machine learning* (Hutter et al., 2019), in particular an efficient *Bayesian optimization* method (Falkner et al., 2018), to address the problems of architecture search (AS) (Zoph & Le, 2017; Elsken et al., 2018) and hyperparameter optimization as a joint optimization problem. To automatically select the best neural architecture based on data, we define a search space that includes both elements of convolutional neural networks (CNNs) and recurrent neural networks (RNNs) and let the optimizer choose the best combination of the two.

In this section, we present our representation of the search space and describe our approach to optimizing performance.

## 4.1 SEARCH SPACE

Our search space has three components described in the following: choices about the policy network's architecture, environment parameters (including the representation of the state and the reward), and training hyperparameters.

**Neural Architecture** We construct the architecture of our policy network as follows: (1) the dot bracket representation of the state is either binary encoded (distinguishing between paired and unpaired sites) or processed by an embedding layer that converts the symbol-based representation into a learnable numerical one for each site. Then, (2) an optional CNN with at most two layers can be selected, followed by (3) an optional LSTM with at most two layers. Finally, we always add (4) a shallow fully-connected network with one or two layers, which outputs the distribution over actions. This parameterization covers a broad range of possible neural architectures while keeping the dimensionality of the search space relatively modest (similar to what is achieved by the focus on cell spaces (Zoph et al., 2018) in the recent literature on architecture search).

**Environment Parameters** Since our ultimate goal is not to solve a specific decision process (DP), but to use the best DP for solving our problem, we also optimize parameters concerning the state representation and the reward: We optimize the number of sites symmetrically centered around the current one via the state radius $\kappa$ (see Section 3.1), and the shape of the reward via the parameter $\alpha$ (see Equation 4).

**Training Hyperparameters** Since the performance of neural networks strongly depends on the training hyperparameters governing optimization and regularization, we optimized some of the parameters of PPO, which we employ for training the network: learning rate, batch size, and strength of the entropy regularization.

Overall, these design choices yield a 14-dimensional search space comprising mostly integer variables. The complete list of parameters, their types, ranges, and the priors we used over them can be found in Appendix E. We used almost identical search spaces for *LEARNA* and *Meta-LEARNA*, but adapted the ranges for the learning rate and the entropy regularization slightly based on preliminary experiments. Please refer to Tables 3 and 4 in Appendix E for more details.

## 4.2 SEARCH PROCEDURE

We now describe how we optimized performance in the search space described above. We chose the recently-proposed optimizer BOHB (Falkner et al., 2018) to find good configurations, because it can handle mixed discrete/continuous spaces, utilize parallel resources, and additionally can exploit cheap approximations of the objective function to speed up the optimization. These so-called *low-fidelity approximations* can be achieved in numerous ways, e.g., limiting the training time, the number

of independent repetitions of the evaluations, or using only fractions of the data. In our setting, we decided to limit the wall-clock time for training (*Meta-LEARNA*) or the evaluations (*LEARNA*). For a detailed description of the limits, we refer to Appendix E.

**Datasets**    To properly optimize the listed design choices without overfitting, we needed a designated training and validation dataset. However, previous benchmarks used in the *RNA Design* literature do not provide a train/validation/test split. This led us to create the benchmark Rfam-Learn based on the Rfam database version 13.0 (Kalvari et al., 2017), by employing the protocol described in Appendix B. All datasets we used for this paper are listed in detail in Appendix D, however, we note that all our approaches were optimized using only our newly introduced training and validation sets (Rfam-Learn-Train and Rfam-Learn-Validation).

**Budgets**    Due to the very different standardized evaluation timeouts of the benchmarks we report on (10 minutes for Rfam-Taneda and up to 24 hours for Eterna100), we experimented with different budgets for *LEARNA*. In particular, we ran our optimization with a 10-minute and a 30-minute evaluation timeout (the former matching the Rfam-Taneda limit, the latter being larger, but still computationally far more manageable than a 24 hour budget per sequence). After the optimization, we evaluated both alternatives on our full validation set with a limit of 1 hour with the following modification that we also used when evaluating on the Eterna100 and Rfam-Learn-Test benchmarks: matching the evaluation timeout during optimization, every 10 or 30 minutes, the policy network and all internal variables of PPO are reinitialized, i.e., we perform a *restart* of the algorithm to overcome occasional stagnation of PPO. We found the 30-minute variant to perform better, and refer to this as *LEARNA* throughout the rest of the paper.

**Objective**    Despite the fact that RL is known to often yield noisy or unreliable outcomes in single optimization runs (Henderson et al., 2017), we actively decided to only use a single optimization run and a single validation set for each configuration to keep the optimization manageable. To counteract the problems associated with single (potentially) noisy observations, we studied three different loss functions for the hyperparameter optimization: (a) The number of unsolved sequences, (b) the sum of mean distances, and (c) the sum of minimum distances to the target structure. While we ultimately seek to minimize (a), this is a rather noisy and discrete quantity. In preliminary experiments, optimizing (b) turned out to be inferior to (c), presumably because the former punishes exploration by the agent more, while the latter rewards ultimately getting close to the solution. Therefore, we used (c) during the optimization, but picked the final configuration using (a) among the top five configurations. All of these evaluations were based on the validation set.

## 5   RELATED WORK

**Architecture and Hyperparameter Search**    Mendoza et al. (2016) and Zela et al. (2018) previously studied joint architecture search and hyperparameter optimization. Here, we adapted this approach for the use in deep RL and to a richer space of architectures. Although RL has been used for performing architecture search (Zoph & Le, 2017; Mortazi & Bagci, 2018) and joint architecture and hyperparameter search (Wong et al., 2018), to the best of our knowledge, this paper is the first application of the reverse: architecture search for RL. For detailed reviews on architecture search and hyperparameter optimization, we refer to Elsken et al. (2018) and Feurer & Hutter (2018), respectively.

**Matter Engineering**    Variational autoencoders, generative adversarial networks and reinforcement learning have recently shown promising results in protein design and other related problems in matter engineering (Gupta & Zou, 2018; Greener et al., 2018; Olivecrona et al., 2017). For a detailed review on machine learning approaches in the field of matter engineering, we refer to Sanchez-Lengeling & Aspuru-Guzik (2018). In recent work related to *RNA Design*, a convolutional neural network based auto-encoder with additional supervised fine tuning was proposed to score on-target and off-target efficacy of *guide RNAs* for the genome editing technique *CRISPR/CAS9* (Chuai et al., 2018). This automated efficacy scoring could inform future endeavours in designing *guide RNAs*. Our work adds evidence for the competitiveness of generative machine learning methods in this general problem domain.

**RNA Design**   Most algorithms targeting the *RNA Design* problem are either local or global algorithms. Local approaches commonly operate on a single sequence and try to find a solution by changing a small number of nucleotides at a time, guided by the loss function (*RNAInverse* (Hofacker et al., 1994), *RNA-SSD* (Andronescu et al., 2004), *INFO-RNA* (Busch & Backofen, 2006), *NUPACK* (Dirks & Pierce, 2004; Zadeh et al., 2010), *ERD* (Esmaili-Taheri et al., 2014) and the approach by Eastman et al. (2018)). Global methods, on the other hand, either have a large number of candidates being manipulated, or model a global distribution from which samples are generated (*MODENA* (Taneda, 2011), *antaRNA* (Kleinkauf et al., 2015) and *MCTS-RNA* (Yang et al., 2017)). A more detailed review can be found in Churkin et al. (2017).

**RNA Design Using Human Solutions**   Very recently, another, less general direction to *RNA Design* imposed a prior of human knowledge onto the agent (Shi et al., 2018). In this approach, a large ensemble of models is trained on human solutions to manually designed *RNA Design* problems. Further, for refinement of the candidate solution, an adaptive walk procedure using human strategies is used, incorporating deep domain-knowledge guiding the agent's behaviour. Totalized results over all models of the ensemble were reported on the Eterna100 benchmark (Anderson-Lee et al., 2016), which solely consists of manually designed *RNA Design* problems, and which we also report on here. Although the approach showed good results in this one benchmark, human solutions and strategies were not available for our further benchmarks derived from natural RNA structures, and due to computational costs we could not include this work in our comparison.

**RL for Combinatorial Problems**   The work by Bello et al. (2016) heavily influenced our work. In it, the authors apply RL to combinatorial problems, namely the *Traveling Salesman Problem*. The agent proposes complete solutions rather than manipulating an existing one, and it is trained using an episodic reward, in this case the negative tour length. Inspired by this work, we propose to frame the *RNA Design* problem as a RL problem where each candidate solution is designed from scratch. In our approach, the agent predicts which nucleotides to place next into the sequence, learning to design RNA end-to-end.

**RL for RNA Design**   Our generative approach is in stark contrast to the recent work Eastman et al. (2018) carried out in parallel to and independently from ours. Eastman et al. used RL to perform a local search starting from a randomly initialized sequence. The RL agent applies local modifications to design a solution that folds into the desired target structure. The current sequence constitutes the state and each action represents changing an unpaired nucleotide or a pair of nucleotides. After each action the current sequence is evaluated utilizing the *Zuker algorithm* (Zuker & Stiegler, 1981; Zuker & Sankoff, 1984) and the agent only receives a nonzero reward signal once it finds a correct sequence. The agent's policy is a convolutional neural network pre-trained on fixed-length, randomly generated sequences. In the remainder of the paper, we refer to this approach as *RL-LS*, since the RL agent performs a local search.

## 6   EXPERIMENTS

We evaluate our approaches against state-of-the-art methods and perform an ablation study to assess the importance of our method's components. We report results on two established benchmarks from the literature and on our own benchmark. Full information on the three benchmarks is given in Appendix D. For each benchmark, we followed its standard evaluation protocol, performing multiple attempts (in the following referred to as evaluation runs) with a fixed time limit for each target structure. For each benchmark, we report the accumulated number of solved targets across all evaluation runs and provide means and standard deviations around the mean for all experiments. All methods were compared on the same hardware, each allowed one CPU core per evaluation of a single target structure. The methods we compare to either do not have clear/exposed hyperparameters (*RNAinverse*), or were optimized by the original authors (*antaRNA*, *RL-LS*, and *MCTS-RNA*); all methods – including our own – might benefit from further optimization of their hyperparameters for specific benchmarks. Details concerning the used software and hardware are listed in Appendix C.

### 6.1   COMPARATIVE STUDY

The results of our comparative study, summarized in Table 1 and Figure 3, are as follows.

Table 1: Fraction of solved target structures for *MCTS-RNA*, *antaRNA*, *RL-LS*, *RNAInverse*, *LEARNA*, *Meta-LEARNA*, and *Meta-LEARNA-Adapt* on the two benchmarks from the literature (Eterna100 and Rfam-Taneda), as well as on our newly introduced benchmark (Rfam-Learn-Test). A target structure counts as solved if a solution was found in any of the evaluation runs.

| METHOD | SOLVED SEQUENCES [%] | | |
|---|---|---|---|
| | ETERNA100 | RFAM-TANEDA | RFAM-LEARN-TEST |
| MCTS-RNA | 57 | 79 | 97 |
| ANTARNA | 58 | 66 | **100** |
| RL-LS | 59 | 62 | 62 |
| RNAINVERSE | 60 | 59 | 95 |
| LEARNA | 67 | 79 | 97 |
| META-LEARNA | **68** | **83** | **100** |
| META-LEARNA-ADAPT | **68** | **83** | 99 |

**Eterna100**    Solving up to 68% (*Meta-LEARNA* and *Meta-LEARNA-Adapt*) of the target structures, all our approaches achieve clear new state-of-the-art results on the Eterna100 benchmark. Additionally, *Meta-LEARNA* only needs about 25 seconds to reach the final performance of any other method ($\approx 1765\times$ faster) and achieves new state-of-the-art results in less than 30 seconds. This performance is stable through all of the five evaluation runs performed. Remarkably, all versions of our approach already achieve new state-of-the-art performance in each single evaluation run (see Appendix I).

**Rfam-Taneda**    Concerning the Rfam-Taneda benchmark, *LEARNA* is on par with the current state-of-the-art results of *MCTS-RNA* after 110 seconds ($\approx 2\times$ faster). *Meta-LEARNA* and *Meta-LEARNA-Adapt* achieve this previous state-of-the-art performance in less than 5 seconds ($\approx 63\times$ faster) and new state-of-the-art results after 400 seconds and 90 seconds, respectively (see Appendix J), solving 83% of the target structures.

**Rfam-Learn**    Only *Meta-LEARNA* and *antaRNA* were able to solve all of the target structures (in 29 minutes and 20 minutes, respectively). Except for *RL-LS*, all algorithms could solve at least 95% of the target structures.

In summary, our novel deep reinforcement learning algorithm achieved the best performance on all of the three benchmarks while being much faster than all other algorithms on the two benchmarks from the literature (Eterna100 and Rfam-Taneda). Our meta-learning approach *Meta-LEARNA* learned a representation of the dynamics underlying *RNA Design* and is capable of transferring this knowledge to new *RNA Design* tasks. As our additional analysis in Appendix H shows, it also scales better with sequence length than existing approaches. For a detailed list of the performance of all algorithms on specific target structures, we refer to the detail tables in Appendix K.

## 6.2    ABLATION STUDY AND PARAMETER IMPORTANCE

To study the influence of the different components and parameters on the performance of our approach, we performed an ablation study and a functional analysis of variance (fANOVA) (Hooker, 2007; Hutter et al., 2014).

**Ablation Study**    For the ablation, we excluded either the adaptation option, the local improvement step, or the restart option. For all variants of our approach we observed a clear boost in performance from the local improvement step, while the other components tended to have a smaller impact (see Figure 8 in Appendix G). We note that we believe the local improvement step could also benefit other generative approaches, such as *MCTS-RNA*. The restart option only boosted performance on the Eterna100 benchmark, with considerably harder instances and a much longer runtime (see Figure 9 in Appendix G). As already apparent from our comparative study (Section 6.1), the continued adaptation (*Meta-LEARNA-Adapt*) of the learned parameters did not improve performance. This might be due to us not having optimized hyperparameters for this variant, but simply having reused the same settings as for *Meta-LEARNA*.

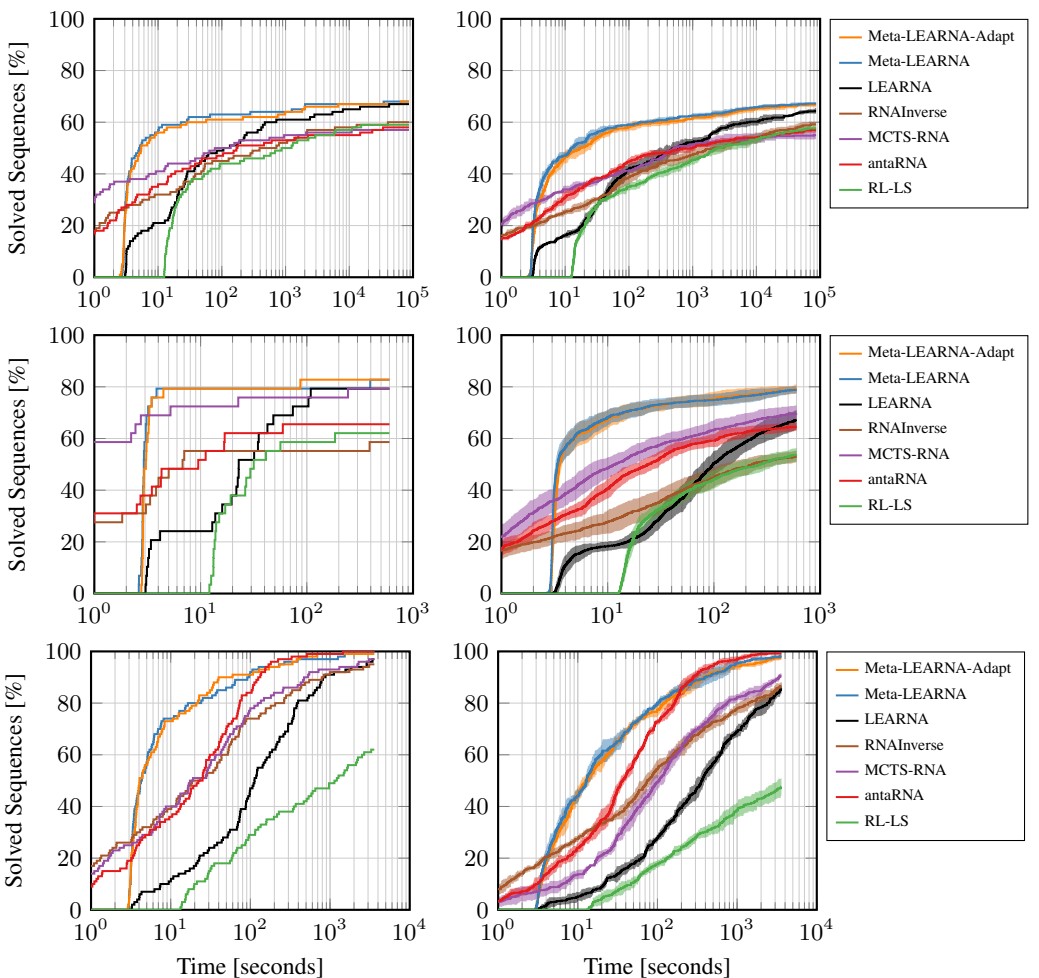

Figure 3: Performance across the time spent on each particular target structure for all methods on the Eterna100 benchmark (top), the Rfam-Taneda benchmark (middle), and the Rfam-Learn-Test benchmark (bottom). On the left we show the total number of target structures that were solved in at least one evaluation run, while the right panels show the average number of solved target structures and the standard deviation around the mean.

**Parameter Importance** The fANOVA results highlight the importance of parameters from all three components of the search space mentioned in Section 4. This emphasizes the importance of the joint optimization of the policy network's architecture, the environment parameters and the training hyperparameters.

All results and a more detailed discussion of our ablation study and the fANOVA results can be found in Appendix G and F, respectively.

# 7 CONCLUSION

We proposed the deep reinforcement learning algorithm *LEARNA* for the *RNA Design* problem to sequentially construct candidate solutions in an end-to-end fashion. By pre-training on a large corpus of biological sequences, a local improvement step to aid the agent, and extensive architecture and hyperparameter optimization, we arrived at *Meta-LEARNA*, a ready-to-use agent that achieves state-of-the-art results on the Eterna100 (Anderson-Lee et al., 2016) and the Rfam-Taneda benchmark (Taneda, 2011). Our ablation study shows the importance of all components, suggesting that RL with an additional local improvement step can solve the *RNA Design* problem efficiently. Code and data for reproducing our results is available at https://github.com/automl/learna.

ACKNOWLEDGMENTS

This work was supported by the European Research Council (ERC) under the European Union's Horizon 2020 research and innovation programme under grant no. 716721, and by the German Research Foundation (DFG), under the BrainLinksBrainTools Cluster of Excellence (grant number EXC 1086). The authors acknowledge support by the state of Baden-Württemberg through bwHPC and the DFG through grant no. INST 39/963-1 FUGG.

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

## A    PSEUDOCODE FOR COMPUTING THE REWARD

---

**Algorithm 1:** Local improvement step (LIS) using Hamming distance $d_H(\cdot, \cdot)$ and folding algorithm $\mathcal{F}(\cdot)$.

---

    **input** : designed solution $\phi$, target structure $\omega$, initial Hamming distance $\delta$
    **output :** locally improved distance
1  $\Delta \leftarrow \emptyset$
2  nucleotide_combinations $\leftarrow \{A, G, U, C\}^{\delta}$
3  candidate_solutions $\leftarrow$ replaceMismatchedSites($\phi$, $\omega$, nucleotide_combinations)
4  **foreach** $\psi \in$ candidate_solutions **do**
5      $\delta \leftarrow d_H(\mathcal{F}(\psi), \omega)$
6      **if** $\delta = 0$ **then**
7         |  **return** $\delta$
8      **end**
9      $\Delta \leftarrow \Delta \cup \{\delta\}$
10 **end**
11 **return** $\min \Delta$

---

**Algorithm 2:** Computing reward $\mathcal{R}_{\omega}^{T}(\phi)$ using LIS (Algorithm 1), Hamming distance $d_H(\cdot, \cdot)$ and folding algorithm $\mathcal{F}(\cdot)$.

---

    **input** : designed solution $\phi$, target structure $\omega$, LIS cut-off parameter $\xi$
    **output :** reward $\mathcal{R}_{\omega}^{T}(\phi)$
1  $\delta \leftarrow d_H(\mathcal{F}(\phi), \omega)$
2  **if** $\delta = 0$ **then**
3     |  **return** $\delta$
4  **else if** $\delta < \xi$ **then**
5     |  $\delta \leftarrow$ LIS($\phi$, $\omega$, $\delta$)
6  **end**
7  $L_{\omega} \leftarrow \delta \,/\, |\omega|$
8  **return** $(1 - L_{\omega})^{\alpha}$

---

## B    CREATING THE RFAM-LEARN DATASETS

To ensure a large enough and interesting dataset, we downloaded all families of the Rfam database version 13.0 (Kalvari et al., 2017) and folded them using the *ViennaRNA* package (Lorenz et al., 2011a). We removed all secondary structures with multiple known solutions, and only kept structures with lengths between 50 and 450 to match the existing datasets. To focus on the harder sequences, we only kept the ones that a single run of *MCTS-RNA* could not solve within 30 seconds. We chose *MCTS-RNA* for filtering as it was the fastest algorithm from the literature. The remaining secondary structures were split into a training set of 65000, a validation set of 100, and a test set of 100 secondary structures.

## C    SOFTWARE AND HARDWARE DETAILS

We used the implementation of the *Zuker algorithm* provided by *ViennaRNA* (Lorenz et al., 2011b) versions 2.4.8 (*MCTS-RNA*, *RL-LS* and *LEARNA*), 2.1.9 (*antaRNA*) and 2.4.9 (*RNAInverse*). Our implementation uses the reinforcement learning library *tensorforce*, version 0.3.3 (Schaarschmidt et al., 2017) working with *TensorFlow* version 1.4.0 (Abadi et al., 2015). All computations were done on Broadwell E5-2630v4 2.2 GHz CPUs with a limitation of 5 GByte RAM per each of the 10 cores. For the training phase of *Meta-LEARNA*, we used two of these CPUs, but at evaluation time, all methods were only allowed a single core (using core binding).

## D  Benchmarks

Table 2: Overview on the three benchmarks Eterna100 (Anderson-Lee et al., 2016), Rfam-Taneda (Taneda, 2011) and Rfam-Learn we used for our experiments. The table displays the timeout, the number of evaluations for each target structure, the number of sequences and the range of sequence lengths for the corresponding benchmark.

| Dataset | Timeout | Evaluations | Sequences | Length |
|---|---|---|---|---|
| Eterna100 | 24h | 5 | 100 | 12−400 |
| Rfam-Taneda | 10min | 50 | 29 | 54−451 |
| Rfam-Learn-Train | − | − | 65000 | 50−450 |
| Rfam-Learn-Val | − | − | 100 | 50−444 |
| Rfam-Learn-Test | 1h | 5 | 100 | 50−446 |

## E  Joint Architecture and Hyperparameter Search

Here, we provide a detailed description of the search space, the different computational budgets used for optimization, and the final configurations found by the optimizer. The search spaces for *LEARNA* and *Meta-LEARNA* can be found in Tables 3 and 4.

Table 3: Search space for the agent's architecture and the hyperparameters used for *LEARNA*.

| Parameter Name | Type | Range | Prior |
|---|---|---|---|
| filter size in $1^{st}$ conv layer | integer | $\{0\} \cup \{3, 5, \ldots, 17\}$ | uniform |
| filter size in $2^{nd}$ conv layer | integer | $\{0, 3, 5, 7, 9\}$ | uniform |
| # filter in $1^{st}$ conv layer | integer | $[1, 32]$ | log-uniform |
| # filter in $2^{nd}$ conv layer | integer | $[1, 32]$ | log-uniform |
| # LSTM layers | integer | $[0, 2]$ | uniform |
| # units in every LSTM layer | integer | $[1, 64]$ | log-uniform |
| # fully connected layers | integer | $[1, 2]$ | uniform |
| # units in fully connected layer(s) | integer | $[8, 64]$ | log-uniform |
| state space radius $\kappa$ | integer | $[0, 32]$ | uniform |
| embedding dimensionality | integer | $[0, 4]$ | uniform |
| batch size | integer | $[32, 128]$ | log-uniform |
| entropy regularization | float | $[1 \cdot 10^{-5},\ 1 \cdot 10^{-2}]$ | log-uniform |
| learning rate for PPO | float | $[1 \cdot 10^{-5},\ 1 \cdot 10^{-3}]$ | log-uniform |
| reward exponent $\alpha$ | float | $[1,\ 10]$ | uniform |

Table 4: Modified hyperparameters in the search space used for optimizing *Meta-LEARNA* compared to Table 3. We adapted these ranges slightly based on preliminary experiments. We hypothesize that the longer training time and the parallel training require smaller learning rates and larger regularization.

| Parameter Name | Type | Range | Prior |
|---|---|---|---|
| entropy regularization | float | $[5 \cdot 10^{-5},\ 5 \cdot 10^{-3}]$ | log-uniform |
| learning rate for PPO | float | $[1 \cdot 10^{-6},\ 1 \cdot 10^{-4}]$ | log-uniform |

Using varying budgets, we can eliminate bad configurations quickly and focus most of the resources on the promising ones. In BOHB, these budgets are geometrically distributed with a factor of three between them. For *LEARNA*, we directly optimize the performance on the validation set and use varying evaluation timeouts as budgets, with a maximum of 30 minutes to keep the optimization manageable. For *Meta-LEARNA*, we vary the training time and keep the evaluation timeout on the validation set fixed at 60 seconds. The maximum timeout of 1 hour on 20 CPU cores was chosen

to almost match the timeout of the Eterna100 benchmark for a single sequence and the minimum timeout was set to 400 seconds, chosen by preliminary runs and inspecting the achieved performance. The validation timeout of one minute was chosen such that the training time on the smallest budget of 400 seconds is still larger than the evaluation time for the 100 validation sequences. Additionally, this encourages the agent to find a solution quickly. These considerations lead to the budgets shown in the legends of Figure 4 and Figure 5.

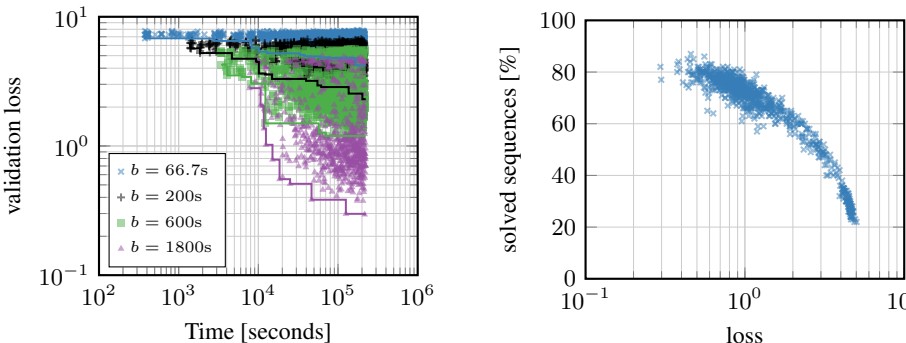

Figure 4: Left: Observed validation loss during the BOHB run for *LEARNA*. The different budgets $b$ correspond to the timeout for each of the 100 validation sequences. Right: Relationship between the observed validation loss (sum of minimal, normalized *Hamming distances*) and the fraction of solved sequences.

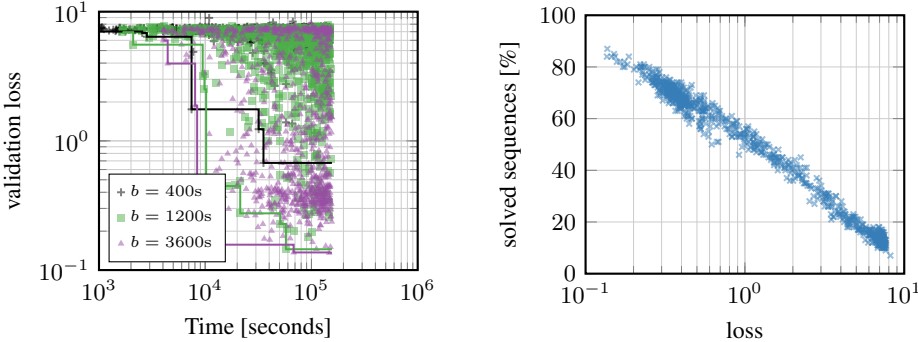

Figure 5: Left: Observed validation loss during the BOHB run for *Meta-LEARNA*. The different budgets $b$ corresponds to the training time on 20 CPU cores before evaluating on the 100 validation sequences for 60 seconds each. The results seem to suggest that one can achieve a very similar performance with only 20 minutes of training, which could imply that much longer training of the agent might be required for substantially better performance. Right: Relationship between the observed validation loss (sum of minimal, normalized *Hamming distances*) and the fraction of solved sequences during validation. The plot suggests that our loss metric correlates strongly with the number of successfully found primary sequences.

Finally, Table 5 summarizes the evaluated configurations. The biggest differences between *LEARNA* and *Meta-LEARNA* can be found among the architectural choices. The *LEARNA* configuration has a relatively big CNN component and additionally uses a single LSTM layer with 28 units; in contrast, the best found *Meta-LEARNA* configuration has no LSTM layers and a relatively small CNN component with only 3 filters in the second layer. For both *LEARNA* and *Meta-LEARNA*, a modest feed forward component with only one layer suffices, the number of embedding dimensions and the batch sizes are almost identical. The entropy regularization and the learning rate also vary, validating our decision to adapt the search spaces based on preliminary experiments. We expect most of these differences to be the result of the different CPU time budgets, but we do not want to speculate about whether CNNs are inherently better suited to generalizing across sequences than LSTMs based on our results; longer training and more optimization might also produce a configuration for *Meta-LEARNA* with LSTM cells.

To summarize the results from the optimization: The best found configurations vary in key parameters, highlighting the necessity to jointly optimize as many aspects of the RL problem as possible for the given scenario.

Table 5: The selected configurations for each scenario.

| Parameter Name | LEARNA | Meta-LEARNA |
|---|---|---|
| filter size in $1^{st}$ conv layer | 17 | 11 |
| filter size in $2^{nd}$ conv layer | 5 | 3 |
| # filters in $1^{st}$ conv layer | 7 | 10 |
| # filters in $2^{nd}$ conv layer | 18 | 3 |
| # fully connected layers | 1 | 1 |
| # units in fully connected layer(s) | 57 | 52 |
| # LSTM layers | 1 | 0 |
| # units in every LSTM layer | 28 | 3 |
| state space radius $\kappa$ | 32 | 29 |
| embedding dimensionality | 3 | 2 |
| batch size | 126 | 123 |
| entropy regularization | $6.76 \cdot 10^{-5}$ | $1.51 \cdot 10^{-4}$ |
| learning rate for PPO | $5.99 \cdot 10^{-4}$ | $6.44 \cdot 10^{-5}$ |
| reward exponent $\alpha$ | 9.34 | 8.93 |

## F  FUNCTIONAL ANALYSIS OF VARIANCE FOR META-LEARNA

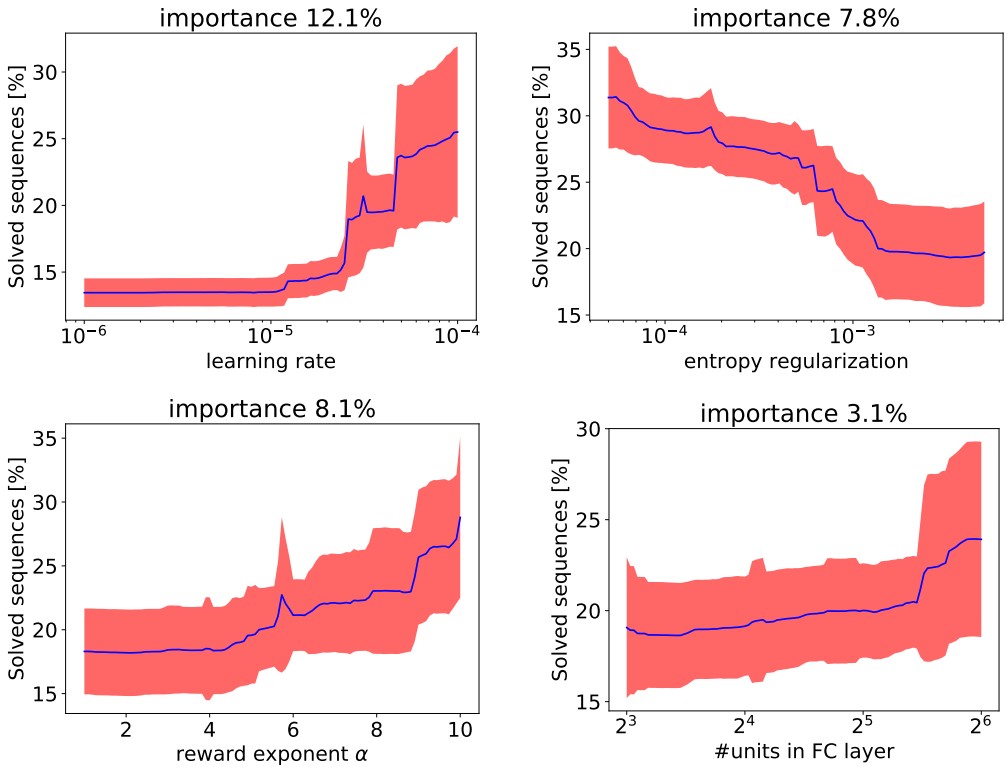

Figure 6: Marginal prediction plots for the most important individual parameters, with all other parameters marginalized out based on a random forest regression model. We plot means of the marginal prediction across the random forest's individual trees $\pm$ the empirical standard deviation across the trees. The importance numbers given in the figure subtitles measure the fraction of the total variance explained by the respective single parameter.

Here, we performed an analysis of variance (ANOVA) that quantifies the global importance of a parameter of *Meta-LEARNA* by the fraction of the total variance it explains. Because our parameter space is rather high dimensional, and we collected a limited (relative to the dimensionality) and highly biased (because we optimized performance) set of evaluations, we use the functional ANOVA (fANOVA) framework (Hooker, 2007). In particular, we use fANOVA based on random forests as introduced by Hutter et al. (2014). The results are shown in Figures 6 and 7.

Among the four most important individual parameters, we found training and regularization hyper-parameters (learning rate and entropy regularization in PPO), the reward representation (the reward exponent), and an architectural hyperparameter (number of units in the fully connected layer(s)). This highlights the need to include all components in the optimization.

The global analysis performed by fANOVA highlights hyperparameters that impact performance most across the entire search space. As a result, the shown fraction of solved validation sequences is rather low in the plots ($\lesssim 35\%$, where the best found configurations achieved almost $90\%$, see Figure 5). It is important to note that the quantitative behavior predicted by the fANOVA does not have to be representative for the best configurations, especially if the *good part* of the space is rather small. This also means that other hyperparameters, e.g., the architecture and type of the network, can be more important than indicated by the fANOVA in order to reach peak performance.

From the plots, we can conclude that a relatively large learning rate performs best on average. Interestingly, it seems to be advantageous to have a limited entropy regularization, which we see as an indicator that the training set is fairly diverse and the problem challenging enough for the agent to

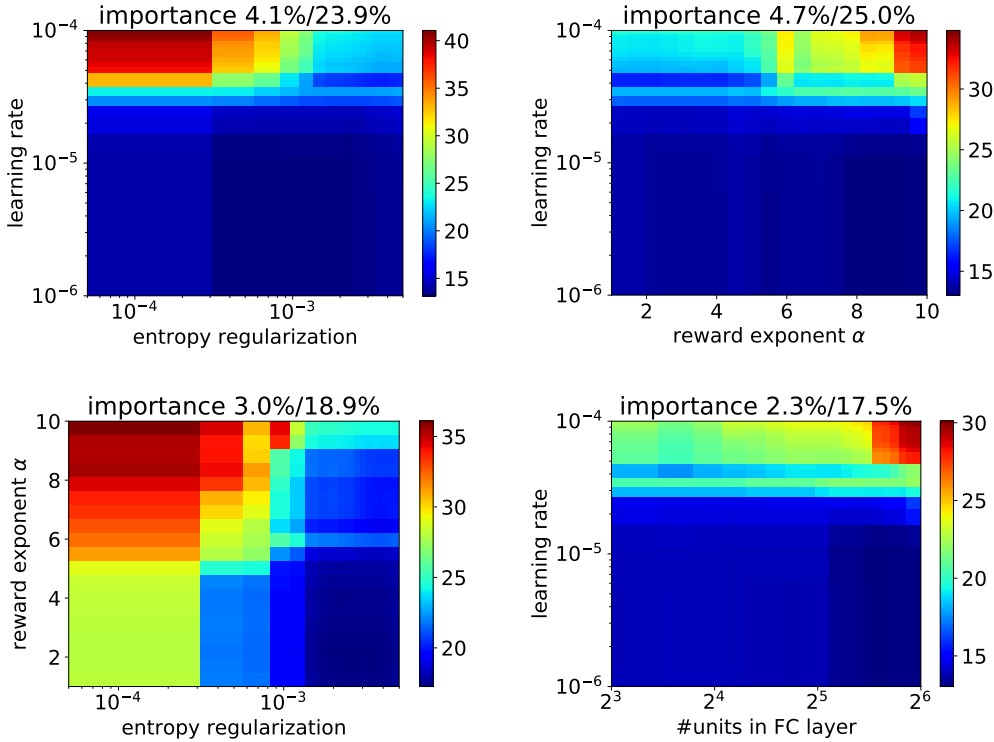

Figure 7: Marginal prediction plots for the most important pairs of parameters when marginalizing across all other parameters. The importance values shown in the subtitles are the ones by the interaction effect itself (first) and the sum of it and the two individual effects (second).

keep exploring. The reward exponent should also be set quite high in conjunction with the learning rate (see top right panel of Figure 7).

# G ABLATION STUDY

In addition to the hyperparameter importance study, we assess the contribution of the different components of our approaches with an ablation (Figure 8 and Figure 9). Clearly, a model based agent compared to random actions has the biggest impact on the performance. The second most important component is the local improvement step, which is active once a sequence with less than 5 mismatches has been found. Restarts only seem to affect the performance on the Eterna100 benchmark, where due to the long budget, we only evaluated *LEARNA*. The seemingly negligible impact of the continued training in *Meta-LEARNA-Adapt* could increase on datasets more dissimilar to the training data or with an additional optimization of the relevant parameters used for the continuous updates. Potentially, all parameters except the architecture and the state space representation could be optimized to improve performance. This could be investigated in future work.

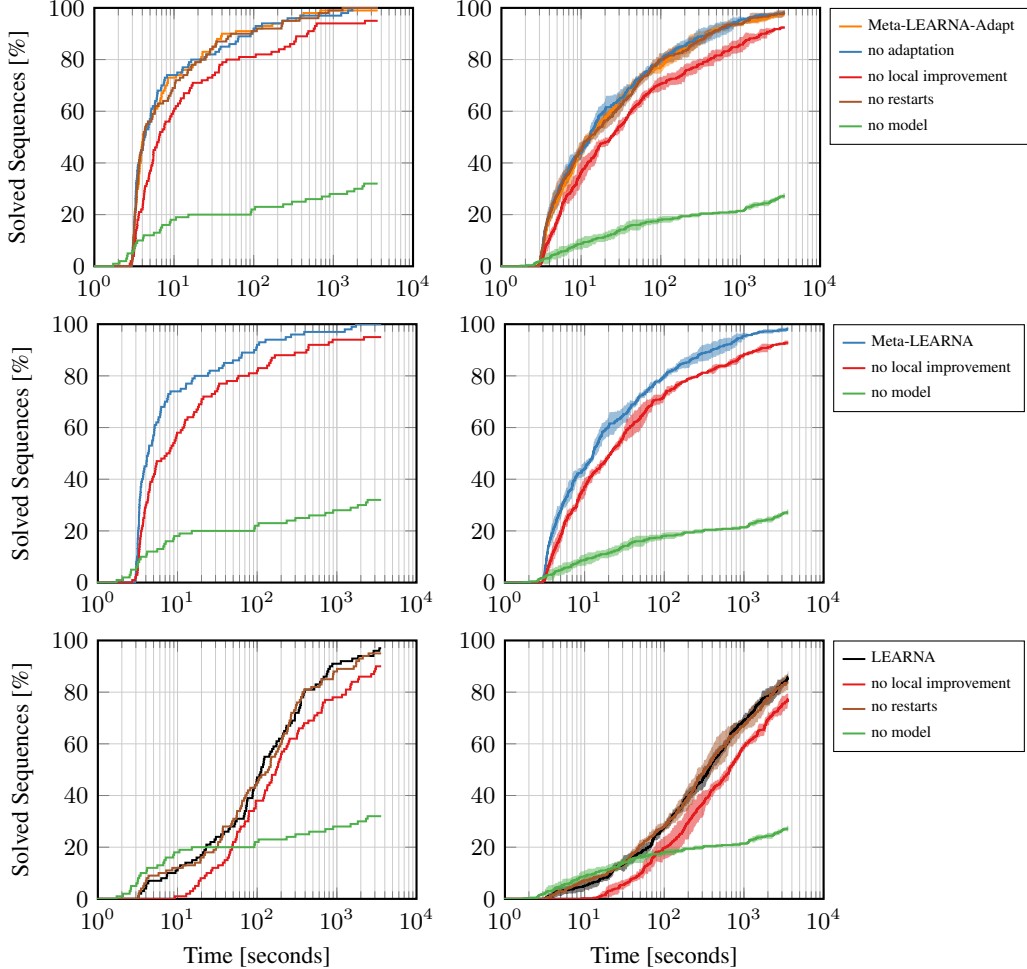

Figure 8: Ablation study of *Meta-LEARNA-Adapt* (first row), *Meta-LEARNA* (second row) and *LEARNA* (third row) on Rfam-Learn-Test. The left side shows the accumulated number of solved target structures over 5 independent runs, while the right side shows the mean and the standard deviation around the mean.

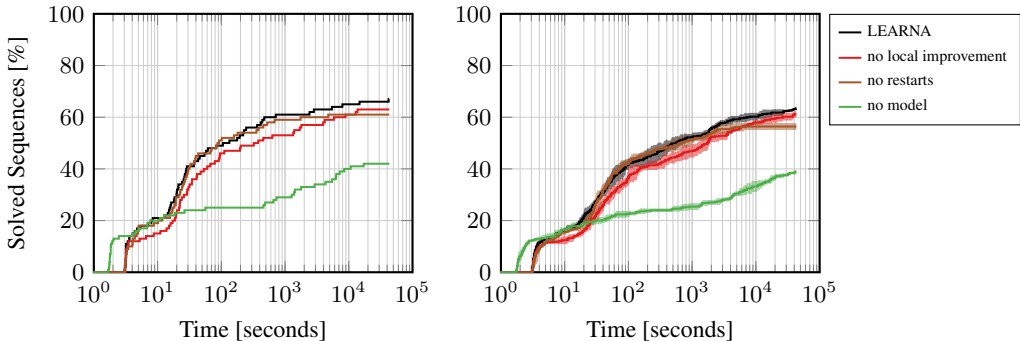

Figure 9: Ablation study of *LEARNA* on Eterna100 with an evaluation timeout of 12 hours. The left side shows the accumulated number of solved target structures over 5 independent runs, while the right side shows the mean and the standard deviation around the mean.

# H   Comparison: Performance Across Sequence Lengths

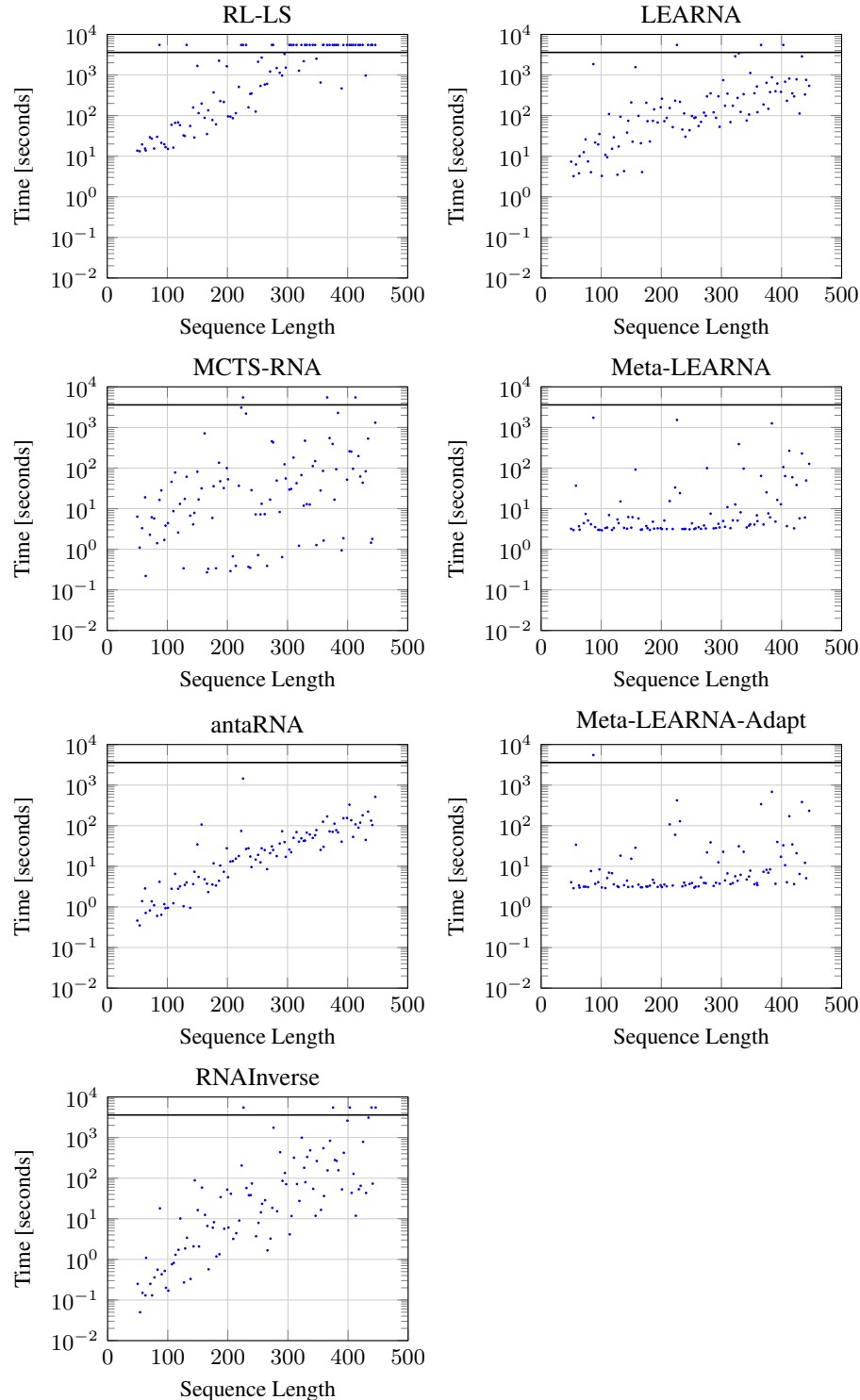

Figure 10: Minimum solution times across sequence lengths on the Rfam-Learn-Test benchmark. The solid line represents the evaluation timeout of 1 hour for the Rfam-Learn-Test benchmark and points drawn above this line were not solved.

# I   COMPARISON: NUMBER OF SOLUTIONS PER k RUNS

Table 6: Comparison of all methods on Eterna100. Results list the number of solved target structures in at least 1, 2, 3, 4, or all of the evaluation runs in percent.

| METHOD | SOLVED SEQUENCES [%] | | | | |
|---|---|---|---|---|---|
| | TOTAL | 2 RUNS | 3 RUNS | 4 RUNS | ALL RUNS |
| MCTS-RNA | 57 | 57 | 56 | 54 | 51 |
| antaRNA | 58 | 58 | 58 | 56 | 55 |
| RL-LS | 59 | 59 | 58 | 57 | 55 |
| RNAINVERSE | 60 | 60 | 59 | 59 | 58 |
| LEARNA | 67 | 66 | 63 | 63 | 63 |
| META-LEARNA | **68** | **67** | **67** | **67** | **67** |
| META-LEARNA-ADAPT | **68** | **67** | **67** | 66 | 66 |

Table 7: Comparison of all methods on Rfam-Taneda. Results list the number of solved target structures in at least 1, 5, 10, 25, or all of the evaluation runs in percent.

| METHOD | SOLVED SEQUENCES [%] | | | | |
|---|---|---|---|---|---|
| | TOTAL | 5 RUNS | 10 RUNS | 25 RUNS | ALL RUNS |
| MCTS-RNA | 79 | 76 | 72 | 72 | 59 |
| antaRNA | 66 | 66 | 66 | 66 | 62 |
| RL-LS | 62 | 62 | 55 | 52 | 48 |
| RNAINVERSE | 59 | 55 | 55 | 52 | 48 |
| LEARNA | 79 | 79 | 76 | 66 | 48 |
| META-LEARNA | **83** | 79 | **79** | **79** | 72 |
| META-LEARNA-ADAPT | **83** | **83** | **79** | **79** | **76** |

Table 8: Comparison of all methods on Rfam-Learn-Test. Results list the number of solved target structures in at least 1, 2, 3, 4, or all of the evaluation runs in percent.

| METHOD | SOLVED SEQUENCES [%] | | | | |
|---|---|---|---|---|---|
| | TOTAL | 2 RUNS | 3 RUNS | 4 RUNS | ALL RUNS |
| MCTS-RNA | 97 | 94 | 91 | 89 | 82 |
| antaRNA | **100** | **99** | **99** | **99** | **99** |
| RL-LS | 62 | 53 | 45 | 41 | 37 |
| RNAINVERSE | 95 | 90 | 87 | 83 | 78 |
| LEARNA | 97 | 93 | 86 | 82 | 71 |
| META-LEARNA | **100** | **99** | 98 | 98 | 96 |
| META-LEARNA-ADAPT | 99 | **99** | **99** | 97 | 94 |

## J COMPARISON: NUMBER OF SOLUTIONS AT DIFFERENT TIMES

Table 9: Comparison of all methods on Eterna100. Results list the number of solved target structures at different time points in percent.

| METHOD | SOLVED SEQUENCES [%] | | | | | | |
|---|---|---|---|---|---|---|---|
| | 10S | 1MIN | 30MIN | 1H | 4H | 12H | 24H |
| MCTS-RNA | 41 | 48 | 55 | 56 | 57 | 57 | 57 |
| ANTARNA | 36 | 46 | 54 | 55 | 55 | 58 | 58 |
| RL-LS | 0 | 40 | 53 | 55 | 58 | 59 | 59 |
| RNAINVERSE | 32 | 44 | 55 | 57 | 59 | 60 | 60 |
| LEARNA | 21 | 47 | 61 | 63 | 65 | 67 | 67 |
| META-LEARNA | **56** | **62** | **65** | **67** | **67** | **68** | **68** |
| META-LEARNA-ADAPT | **56** | 61 | 64 | 66 | **67** | 67 | **68** |

Table 10: Comparison of all methods on Rfam-Taneda. Results list the number of solved target structures at different time points in percent.

| METHOD | SOLVED SEQUENCES [%] | | | | |
|---|---|---|---|---|---|
| | 10S | 30S | 1MIN | 5MIN | 10MIN |
| MCTS-RNA | 72 | 76 | 76 | 79 | 79 |
| ANTARNA | 52 | 62 | 66 | 66 | 66 |
| RL-LS | 0 | 48 | 59 | 62 | 62 |
| RNAINVERSE | 55 | 55 | 55 | 55 | 59 |
| LEARNA | 24 | 52 | 69 | 79 | 79 |
| META-LEARNA | **79** | **79** | **79** | 79 | **83** |
| META-LEARNA-ADAPT | **79** | **79** | **79** | **83** | **83** |

Table 11: Comparison of all methods on Rfam-Learn-Test. Results list the number of solved target structures at different time points in percent.

| METHOD | SOLVED SEQUENCES [%] | | | | | | |
|---|---|---|---|---|---|---|---|
| | 10S | 30S | 1MIN | 5MIN | 10MIN | 30MIN | 1H |
| MCTS-RNA | 40 | 55 | 68 | 86 | 92 | 94 | 97 |
| ANTARNA | 36 | 58 | 73 | **97** | **99** | **100** | **100** |
| RL-LS | 0 | 14 | 21 | 38 | 45 | 56 | 62 |
| RNAINVERSE | 39 | 53 | 66 | 83 | 89 | 93 | 95 |
| LEARNA | 11 | 23 | 31 | 72 | 83 | 93 | 97 |
| META-LEARNA | **74** | 82 | 87 | 96 | 97 | **100** | **100** |
| META-LEARNA-ADAPT | 73 | **84** | **91** | 95 | 98 | 99 | 99 |

# K    COMPARISON: SPECIFIC TARGET STRUCTURES

Table 12: Results for 5 independent attempts on the first half of the 100 target structures of the Rfam-Learn-Test benchmark. We abbreviate *Meta-LEARNA* with *M-LEARNA* and *Meta-LEARNA-Adapt* with *M-LEARNA-A*.

| ID | LEARNA | M-LEARNA | M-LEARNA-A | MCTS-RNA | RL-LS | RNAINVERSE | ANTARNA |
|---|---|---|---|---|---|---|---|
| 1 | 5/5 | 5/5 | 5/5 | 5/5 | 5/5 | 5/5 | 5/5 |
| 2 | 5/5 | 5/5 | 5/5 | 5/5 | 5/5 | 5/5 | 5/5 |
| 3 | 5/5 | 5/5 | 5/5 | 5/5 | 5/5 | 5/5 | 5/5 |
| 4 | 5/5 | 5/5 | 5/5 | 5/5 | 5/5 | 5/5 | 5/5 |
| 5 | 5/5 | 5/5 | 5/5 | 5/5 | 5/5 | 5/5 | 5/5 |
| 6 | 5/5 | 5/5 | 5/5 | 5/5 | 5/5 | 5/5 | 5/5 |
| 7 | 5/5 | 5/5 | 5/5 | 5/5 | 5/5 | 5/5 | 5/5 |
| 8 | 5/5 | 5/5 | 5/5 | 5/5 | 5/5 | 5/5 | 5/5 |
| 9 | 5/5 | 5/5 | 5/5 | 5/5 | 5/5 | 5/5 | 5/5 |
| 10 | 2/5 | 2/5 | - | 4/5 | - | 5/5 | 5/5 |
| 11 | 5/5 | 5/5 | 5/5 | 5/5 | 5/5 | 5/5 | 5/5 |
| 12 | 5/5 | 5/5 | 5/5 | 5/5 | 5/5 | 5/5 | 5/5 |
| 13 | 5/5 | 5/5 | 5/5 | 5/5 | 5/5 | 5/5 | 5/5 |
| 14 | 5/5 | 5/5 | 5/5 | 5/5 | 5/5 | 5/5 | 5/5 |
| 15 | 5/5 | 5/5 | 5/5 | 5/5 | 5/5 | 5/5 | 5/5 |
| 16 | 5/5 | 5/5 | 5/5 | 5/5 | 5/5 | 5/5 | 5/5 |
| 17 | 5/5 | 5/5 | 5/5 | 5/5 | 5/5 | 5/5 | 5/5 |
| 18 | 5/5 | 5/5 | 5/5 | 5/5 | 5/5 | 5/5 | 5/5 |
| 19 | 5/5 | 5/5 | 5/5 | 5/5 | 5/5 | 5/5 | 5/5 |
| 20 | 5/5 | 5/5 | 5/5 | 5/5 | 5/5 | 5/5 | 5/5 |
| 21 | 5/5 | 5/5 | 5/5 | 5/5 | 5/5 | 5/5 | 5/5 |
| 22 | 5/5 | 5/5 | 5/5 | 5/5 | - | 5/5 | 5/5 |
| 23 | 5/5 | 5/5 | 5/5 | 5/5 | 5/5 | 5/5 | 5/5 |
| 24 | 5/5 | 5/5 | 5/5 | 5/5 | 5/5 | 5/5 | 5/5 |
| 25 | 5/5 | 5/5 | 5/5 | 5/5 | 4/5 | 5/5 | 5/5 |
| 26 | 4/5 | 5/5 | 5/5 | 4/5 | 2/5 | 5/5 | 5/5 |
| 27 | 5/5 | 5/5 | 5/5 | 5/5 | 5/5 | 5/5 | 5/5 |
| 28 | 2/5 | 5/5 | 5/5 | 5/5 | 1/5 | 5/5 | 5/5 |
| 29 | 5/5 | 5/5 | 5/5 | 3/5 | 3/5 | 5/5 | 5/5 |
| 30 | 5/5 | 5/5 | 5/5 | 5/5 | 5/5 | 5/5 | 5/5 |
| 31 | 5/5 | 5/5 | 5/5 | 5/5 | 5/5 | 5/5 | 5/5 |
| 32 | 5/5 | 5/5 | 5/5 | 5/5 | 5/5 | 5/5 | 5/5 |
| 33 | 5/5 | 5/5 | 5/5 | 5/5 | 5/5 | 5/5 | 5/5 |
| 34 | 5/5 | 5/5 | 5/5 | 5/5 | 5/5 | 5/5 | 5/5 |
| 35 | 5/5 | 5/5 | 5/5 | 5/5 | 4/5 | 5/5 | 5/5 |
| 36 | 5/5 | 5/5 | 5/5 | 4/5 | 3/5 | 5/5 | 5/5 |
| 37 | 5/5 | 5/5 | 5/5 | 5/5 | 5/5 | 5/5 | 5/5 |
| 38 | 5/5 | 5/5 | 5/5 | 5/5 | 1/5 | 5/5 | 5/5 |
| 39 | 5/5 | 5/5 | 5/5 | 5/5 | 5/5 | 5/5 | 5/5 |
| 40 | 5/5 | 5/5 | 5/5 | 5/5 | 5/5 | 5/5 | 5/5 |
| 41 | 5/5 | 5/5 | 5/5 | 5/5 | 5/5 | 5/5 | 5/5 |
| 42 | 5/5 | 5/5 | 5/5 | 5/5 | 5/5 | 5/5 | 5/5 |
| 43 | 5/5 | 5/5 | 5/5 | 5/5 | 5/5 | 5/5 | 5/5 |
| 44 | 2/5 | 5/5 | 5/5 | 1/5 | - | 4/5 | 5/5 |
| 45 | - | 1/5 | 3/5 | - | - | - | 1/5 |
| 46 | 5/5 | 5/5 | 5/5 | 1/5 | - | 5/5 | 5/5 |
| 47 | 5/5 | 5/5 | 5/5 | 5/5 | 3/5 | 5/5 | 5/5 |
| 48 | 5/5 | 5/5 | 5/5 | 5/5 | 5/5 | 5/5 | 5/5 |
| 49 | 5/5 | 5/5 | 5/5 | 5/5 | 5/5 | 5/5 | 5/5 |
| 50 | 5/5 | 5/5 | 5/5 | 5/5 | 4/5 | 5/5 | 5/5 |
| TOTAL | 235/250 | 243/250 | 243/250 | 232/250 | 205/250 | 244/250 | 246/250 |
| SOLVED | 49/50 | 50/50 | 49/50 | 49/50 | 45/50 | 49/50 | 50/50 |

Table 13: Results for 5 independent attempts on the second half of the 100 target structures of the Rfam-Learn-Test benchmark. We abreviate *Meta-LEARNA* with *M-LEARNA* and *Meta-LEARNA-Adapt* with *M-LEARNA-A*.

| ID | LEARNA | M-LEARNA | M-LEARNA-A | MCTS-RNA | RL-LS | RNAInverse | antaRNA |
|----|--------|----------|------------|----------|-------|------------|---------|
| 51 | 5/5 | 5/5 | 5/5 | 5/5 | 1/5 | 5/5 | 5/5 |
| 52 | 5/5 | 5/5 | 5/5 | 5/5 | 5/5 | 5/5 | 5/5 |
| 53 | 5/5 | 5/5 | 5/5 | 5/5 | 1/5 | 5/5 | 5/5 |
| 54 | 5/5 | 5/5 | 5/5 | 5/5 | 2/5 | 5/5 | 5/5 |
| 55 | 5/5 | 5/5 | 5/5 | 5/5 | 3/5 | 5/5 | 5/5 |
| 56 | 4/5 | 5/5 | 5/5 | 5/5 | 1/5 | 5/5 | 5/5 |
| 57 | 5/5 | 5/5 | 5/5 | 5/5 | - | 4/5 | 5/5 |
| 58 | 5/5 | 5/5 | 5/5 | 5/5 | - | 3/5 | 5/5 |
| 59 | 5/5 | 5/5 | 5/5 | 5/5 | 2/5 | 5/5 | 5/5 |
| 60 | 5/5 | 5/5 | 5/5 | 5/5 | 2/5 | 3/5 | 5/5 |
| 61 | 5/5 | 5/5 | 5/5 | 4/5 | 2/5 | 1/5 | 5/5 |
| 62 | 3/5 | 5/5 | 5/5 | 5/5 | 1/5 | 5/5 | 5/5 |
| 63 | 4/5 | 5/5 | 5/5 | 5/5 | 1/5 | 5/5 | 5/5 |
| 64 | 5/5 | 5/5 | 5/5 | 5/5 | - | 5/5 | 5/5 |
| 65 | 5/5 | 5/5 | 5/5 | 5/5 | - | 5/5 | 5/5 |
| 66 | 4/5 | 5/5 | 5/5 | 5/5 | - | 5/5 | 5/5 |
| 67 | 5/5 | 5/5 | 5/5 | 5/5 | - | 5/5 | 5/5 |
| 68 | 5/5 | 5/5 | 5/5 | 5/5 | 2/5 | 5/5 | 5/5 |
| 69 | 2/5 | 5/5 | 5/5 | 5/5 | - | 3/5 | 5/5 |
| 70 | 5/5 | 5/5 | 5/5 | 5/5 | 1/5 | 5/5 | 5/5 |
| 71 | 1/5 | 5/5 | 5/5 | 5/5 | - | 5/5 | 5/5 |
| 72 | 5/5 | 5/5 | 5/5 | 5/5 | - | 5/5 | 5/5 |
| 73 | 5/5 | 5/5 | 5/5 | 5/5 | - | 5/5 | 5/5 |
| 74 | 5/5 | 5/5 | 5/5 | 5/5 | - | 2/5 | 5/5 |
| 75 | 5/5 | 5/5 | 5/5 | 5/5 | - | 5/5 | 5/5 |
| 76 | 4/5 | 5/5 | 5/5 | 5/5 | 1/5 | 2/5 | 5/5 |
| 77 | 5/5 | 5/5 | 5/5 | 5/5 | 4/5 | 5/5 | 5/5 |
| 78 | 5/5 | 5/5 | 5/5 | 5/5 | - | 4/5 | 5/5 |
| 79 | 4/5 | 5/5 | 5/5 | 5/5 | - | 5/5 | 5/5 |
| 80 | - | 4/5 | 5/5 | - | - | 1/5 | 5/5 |
| 81 | 3/5 | 5/5 | 5/5 | 5/5 | - | 2/5 | 5/5 |
| 82 | 4/5 | 5/5 | 5/5 | 4/5 | - | - | 5/5 |
| 83 | 4/5 | 5/5 | 5/5 | 5/5 | - | 4/5 | 5/5 |
| 84 | 5/5 | 5/5 | 5/5 | 4/5 | - | 3/5 | 5/5 |
| 85 | 2/5 | 4/5 | 3/5 | 1/5 | - | 5/5 | 5/5 |
| 86 | 5/5 | 5/5 | 5/5 | 5/5 | 2/5 | 5/5 | 5/5 |
| 87 | 3/5 | 5/5 | 5/5 | 5/5 | - | 5/5 | 5/5 |
| 88 | 5/5 | 5/5 | 5/5 | 5/5 | - | 1/5 | 5/5 |
| 89 | - | 5/5 | 5/5 | 2/5 | - | - | 5/5 |
| 90 | 2/5 | 5/5 | 5/5 | 2/5 | - | 5/5 | 5/5 |
| 91 | 5/5 | 5/5 | 5/5 | 5/5 | - | 5/5 | 5/5 |
| 92 | 3/5 | 5/5 | 4/5 | - | - | 5/5 | 5/5 |
| 93 | 1/5 | 5/5 | 5/5 | 4/5 | - | 4/5 | 5/5 |
| 94 | 5/5 | 5/5 | 5/5 | 5/5 | - | 5/5 | 5/5 |
| 95 | 2/5 | 5/5 | 5/5 | 5/5 | - | 1/5 | 5/5 |
| 96 | 4/5 | 5/5 | 5/5 | 5/5 | 2/5 | 5/5 | 5/5 |
| 97 | 1/5 | 5/5 | 4/5 | 2/5 | - | 1/5 | 5/5 |
| 98 | 4/5 | 5/5 | 5/5 | 5/5 | - | - | 5/5 |
| 99 | 4/5 | 5/5 | 5/5 | 5/5 | - | 5/5 | 5/5 |
| 100 | 1/5 | 5/5 | 4/5 | 3/5 | - | - | 5/5 |
| TOTAL | 194/250 | 248/250 | 246/250 | 221/250 | 33/250 | 189/250 | 250/250 |
| SOLVED | 48/50 | 50/50 | 50/50 | 48/50 | 17/50 | 46/50 | 50/50 |

Table 14: Results for 5 independent attempts on the first half of the 100 target structures of the Eterna100 benchmark. We abbreviate *Meta-LEARNA* with *M-LEARNA* and *Meta-LEARNA-Adapt* with *M-LEARNA-A*.

| ID | LEARNA | M-LEARNA | M-LEARNA-A | MCTS-RNA | RL-LS | RNAInverse | antaRNA |
|----|--------|----------|------------|----------|-------|------------|---------|
| 1 | 5/5 | 5/5 | 5/5 | 5/5 | 5/5 | 5/5 | 5/5 |
| 2 | 5/5 | 5/5 | 5/5 | 5/5 | 5/5 | 5/5 | 5/5 |
| 3 | 5/5 | 5/5 | 5/5 | 5/5 | 5/5 | 5/5 | 5/5 |
| 4 | 5/5 | 5/5 | 5/5 | 5/5 | 5/5 | 5/5 | 5/5 |
| 5 | 5/5 | 5/5 | 5/5 | 5/5 | 5/5 | 5/5 | 5/5 |
| 6 | 5/5 | 5/5 | 5/5 | 5/5 | - | 5/5 | 5/5 |
| 7 | 5/5 | 5/5 | 5/5 | 5/5 | 5/5 | 5/5 | 5/5 |
| 8 | 5/5 | 5/5 | 5/5 | 5/5 | 5/5 | 5/5 | 5/5 |
| 9 | 5/5 | 5/5 | 5/5 | - | 4/5 | - | 3/5 |
| 10 | 5/5 | 5/5 | 5/5 | - | 5/5 | 5/5 | 5/5 |
| 11 | 5/5 | 5/5 | 5/5 | 5/5 | 5/5 | 5/5 | 5/5 |
| 12 | 5/5 | 5/5 | 5/5 | 5/5 | 5/5 | 5/5 | 5/5 |
| 13 | 5/5 | 5/5 | 5/5 | 5/5 | 5/5 | 5/5 | 5/5 |
| 14 | 5/5 | 5/5 | 5/5 | 5/5 | 5/5 | 5/5 | 5/5 |
| 15 | 5/5 | 5/5 | 5/5 | 5/5 | 5/5 | 5/5 | 5/5 |
| 16 | 5/5 | 5/5 | 5/5 | - | 3/5 | - | - |
| 17 | 5/5 | 5/5 | 5/5 | 4/5 | 5/5 | 2/5 | - |
| 18 | 5/5 | 5/5 | 5/5 | 5/5 | 5/5 | 5/5 | 5/5 |
| 19 | 5/5 | 5/5 | 5/5 | 5/5 | 5/5 | 5/5 | 5/5 |
| 20 | 5/5 | 5/5 | 5/5 | 4/5 | 5/5 | 5/5 | 5/5 |
| 21 | 5/5 | 5/5 | 5/5 | 5/5 | 5/5 | 5/5 | 5/5 |
| 22 | 2/5 | 5/5 | 5/5 | 5/5 | 5/5 | - | 5/5 |
| 23 | 5/5 | 5/5 | 5/5 | - | 5/5 | 5/5 | 5/5 |
| 24 | 5/5 | 5/5 | 5/5 | 5/5 | 5/5 | 5/5 | 5/5 |
| 25 | 5/5 | 5/5 | 5/5 | 5/5 | 5/5 | 5/5 | 5/5 |
| 26 | 5/5 | 5/5 | 5/5 | 5/5 | 5/5 | 5/5 | 5/5 |
| 27 | 5/5 | 5/5 | 5/5 | 5/5 | 5/5 | 5/5 | 5/5 |
| 28 | 5/5 | 5/5 | 5/5 | 5/5 | 5/5 | 5/5 | 5/5 |
| 29 | 5/5 | 5/5 | 5/5 | 5/5 | 5/5 | 5/5 | 5/5 |
| 30 | 5/5 | 5/5 | 5/5 | - | 5/5 | 5/5 | 5/5 |
| 31 | 5/5 | 5/5 | 5/5 | 5/5 | 5/5 | 5/5 | 5/5 |
| 32 | 5/5 | 5/5 | 5/5 | 5/5 | 5/5 | 5/5 | 5/5 |
| 33 | 5/5 | 5/5 | 5/5 | - | - | 5/5 | 5/5 |
| 34 | 5/5 | 5/5 | 5/5 | 5/5 | 5/5 | 5/5 | 5/5 |
| 35 | 2/5 | 5/5 | 5/5 | - | - | - | - |
| 36 | 5/5 | 5/5 | 5/5 | 5/5 | 5/5 | 5/5 | 5/5 |
| 37 | 5/5 | 5/5 | 5/5 | 5/5 | 2/5 | 4/5 | - |
| 38 | 5/5 | 5/5 | 5/5 | 5/5 | - | - | - |
| 39 | 5/5 | 5/5 | 5/5 | 5/5 | 5/5 | 5/5 | 5/5 |
| 40 | 5/5 | 5/5 | 5/5 | 5/5 | 5/5 | 5/5 | 5/5 |
| 41 | 5/5 | 5/5 | 5/5 | - | 5/5 | 5/5 | 5/5 |
| 42 | 5/5 | 5/5 | 5/5 | 4/5 | 5/5 | 5/5 | 5/5 |
| 43 | 5/5 | 5/5 | 5/5 | 5/5 | 5/5 | 5/5 | 5/5 |
| 44 | 5/5 | 5/5 | 5/5 | 5/5 | 5/5 | 5/5 | 5/5 |
| 45 | 5/5 | 5/5 | 5/5 | 5/5 | 5/5 | 5/5 | 5/5 |
| 46 | 5/5 | 5/5 | 5/5 | 5/5 | 5/5 | 5/5 | 5/5 |
| 47 | 5/5 | 5/5 | 5/5 | 5/5 | 5/5 | 5/5 | 5/5 |
| 48 | 5/5 | 5/5 | 5/5 | 5/5 | 5/5 | 5/5 | 5/5 |
| 49 | 5/5 | 5/5 | 5/5 | 5/5 | 5/5 | 5/5 | 5/5 |
| 50 | - | - | - | - | - | - | - |
| TOTAL | 239/250 | 245/250 | 245/250 | 202/250 | 219/250 | 216/250 | 218/250 |
| SOLVED | 49/50 | 49/50 | 49/50 | 41/50 | 45/50 | 44/50 | 44/50 |

Table 15: Results for 5 independent attempts on the second half of the 100 target structures of the Eterna100 benchmark. We abbreviate *Meta-LEARNA* with *M-LEARNA* and *Meta-LEARNA-Adapt* with *M-LEARNA-A*.

| ID | LEARNA | M-LEARNA | M-LEARNA-A | MCTS-RNA | RL-LS | RNAInverse | antaRNA |
|---|---|---|---|---|---|---|---|
| 51 | 5/5 | 5/5 | 5/5 | 5/5 | 5/5 | 5/5 | 5/5 |
| 52 | - | - | - | - | - | - | - |
| 53 | - | 5/5 | 5/5 | - | - | - | - |
| 54 | 5/5 | 5/5 | 5/5 | 5/5 | 5/5 | 5/5 | 5/5 |
| 55 | 5/5 | 5/5 | 5/5 | 5/5 | 5/5 | 5/5 | 5/5 |
| 56 | 5/5 | 5/5 | 5/5 | 5/5 | 5/5 | 5/5 | 5/5 |
| 57 | - | - | - | - | - | - | - |
| 58 | 5/5 | 5/5 | 5/5 | 5/5 | 5/5 | 5/5 | 5/5 |
| 59 | 5/5 | 5/5 | 5/5 | 5/5 | 5/5 | 5/5 | - |
| 60 | - | - | - | - | - | - | - |
| 61 | - | - | - | - | - | - | - |
| 62 | 5/5 | 5/5 | 5/5 | - | - | 5/5 | 3/5 |
| 63 | 5/5 | 5/5 | 5/5 | 5/5 | 5/5 | 5/5 | 5/5 |
| 64 | - | - | - | - | - | - | - |
| 65 | - | - | - | 2/5 | - | - | - |
| 66 | - | 5/5 | - | - | - | 5/5 | 5/5 |
| 67 | - | - | - | - | - | - | - |
| 68 | - | - | - | - | - | - | - |
| 69 | 5/5 | 5/5 | 5/5 | 5/5 | - | - | - |
| 70 | 5/5 | 5/5 | 5/5 | 3/5 | - | - | 4/5 |
| 71 | - | - | - | - | - | - | - |
| 72 | - | - | - | - | 5/5 | 5/5 | - |
| 73 | - | - | - | - | - | - | - |
| 74 | 1/5 | - | 3/5 | - | - | - | - |
| 75 | 5/5 | 5/5 | 5/5 | 5/5 | 5/5 | 5/5 | 5/5 |
| 76 | - | - | - | - | - | - | - |
| 77 | 2/5 | 5/5 | 5/5 | 3/5 | 4/5 | 5/5 | - |
| 78 | - | - | - | - | - | - | - |
| 79 | - | - | - | - | - | - | - |
| 80 | - | - | - | - | - | - | - |
| 81 | - | - | - | - | - | - | - |
| 82 | 5/5 | 5/5 | 5/5 | 5/5 | 5/5 | 5/5 | 5/5 |
| 83 | - | - | - | - | - | - | - |
| 84 | 5/5 | 5/5 | 5/5 | 5/5 | 5/5 | 5/5 | 5/5 |
| 85 | - | - | - | - | - | - | - |
| 86 | - | - | - | - | - | - | - |
| 87 | - | - | - | - | - | - | - |
| 88 | - | - | - | - | - | - | - |
| 89 | - | - | - | - | - | - | - |
| 90 | - | - | - | - | - | - | - |
| 91 | - | - | - | - | - | - | - |
| 92 | - | - | - | - | - | - | - |
| 93 | 5/5 | 5/5 | 5/5 | 5/5 | 5/5 | 5/5 | 5/5 |
| 94 | - | - | - | - | - | - | - |
| 95 | 5/5 | 5/5 | 5/5 | 5/5 | 5/5 | 5/5 | 5/5 |
| 96 | - | - | - | - | - | - | - |
| 97 | - | - | - | - | - | - | - |
| 98 | 5/5 | 1/5 | 1/5 | - | - | - | - |
| 99 | - | - | - | - | - | - | - |
| 100 | - | - | - | - | - | - | - |
| TOTAL | 83/250 | 91/250 | 89/250 | 73/250 | 69/250 | 80/250 | 67/250 |
| SOLVED | 18/50 | 19/50 | 19/50 | 16/50 | 14/50 | 16/50 | 14/50 |

Table 16: Results for 50 independent attempts on each of the 29 target structures of the Rfam-Taneda benchmark. We abbreviate *Meta-LEARNA* with *M-LEARNA* and *Meta-LEARNA-Adapt* with *M-LEARNA-A*.

| ID | LEARNA | M-LEARNA | M-LEARNA-A | MCTS-RNA | RL-LS | RNAInverse | antaRNA |
|---|---|---|---|---|---|---|---|
| 1 | 50/50 | 50/50 | 50/50 | 32/50 | 7/50 | 20/50 | 50/50 |
| 2 | 35/50 | 50/50 | 50/50 | 28/50 | 5/50 | - | - |
| 3 | 18/50 | 49/50 | 50/50 | 4/50 | - | - | - |
| 4 | 50/50 | 50/50 | 50/50 | 50/50 | 50/50 | 50/50 | 50/50 |
| 5 | 50/50 | 50/50 | 50/50 | 50/50 | 50/50 | 50/50 | 50/50 |
| 6 | 50/50 | 50/50 | 50/50 | 50/50 | 50/50 | 50/50 | 50/50 |
| 7 | 50/50 | 50/50 | 50/50 | 50/50 | 48/50 | 50/50 | 50/50 |
| 8 | 50/50 | 50/50 | 50/50 | 50/50 | 50/50 | 50/50 | 50/50 |
| 9 | 18/50 | 50/50 | 50/50 | 44/50 | - | - | - |
| 10 | - | - | - | - | - | - | - |
| 11 | - | - | - | - | - | - | - |
| 12 | 48/50 | 50/50 | 50/50 | 50/50 | 50/50 | 3/50 | 50/50 |
| 13 | 50/50 | 50/50 | 50/50 | 50/50 | 50/50 | 50/50 | 50/50 |
| 14 | 50/50 | 50/50 | 50/50 | 50/50 | 50/50 | 50/50 | 50/50 |
| 15 | 50/50 | 50/50 | 50/50 | 50/50 | 50/50 | 50/50 | 50/50 |
| 16 | - | - | - | - | - | - | - |
| 17 | 22/50 | 50/50 | 50/50 | 47/50 | - | 50/50 | 50/50 |
| 18 | - | 2/50 | 5/50 | - | - | - | - |
| 19 | 50/50 | 50/50 | 50/50 | 50/50 | 50/50 | 50/50 | 50/50 |
| 20 | - | - | - | - | - | - | - |
| 21 | 50/50 | 50/50 | 50/50 | 50/50 | 50/50 | 50/50 | 50/50 |
| 22 | 50/50 | 50/50 | 50/50 | 50/50 | 50/50 | 50/50 | 50/50 |
| 23 | - | - | - | - | - | - | - |
| 24 | 48/50 | 50/50 | 50/50 | 50/50 | 19/50 | - | 50/50 |
| 25 | 50/50 | 50/50 | 50/50 | 50/50 | 50/50 | 46/50 | 50/50 |
| 26 | 50/50 | 50/50 | 50/50 | 50/50 | 50/50 | 50/50 | 50/50 |
| 27 | 8/50 | 42/50 | 43/50 | 6/50 | - | - | - |
| 28 | 49/50 | 50/50 | 50/50 | 50/50 | 50/50 | 50/50 | 50/50 |
| 29 | 28/50 | 50/50 | 50/50 | 50/50 | - | - | 36/50 |
| TOTAL | 974/1450 | 1143/1450 | 1148/1450 | 1011/1450 | 779/1450 | 769/1450 | 936/1450 |
| SOLVED | 23/29 | 24/29 | 24/29 | 23/29 | 18/29 | 17/29 | 19/29 |

