# OpenReview forum: "Learning to Design RNA"
_ICLR.cc/2019/Conference_

### Official Review · AnonReviewer1 · 2018-10-26
**Partially unclear and minor methodological contributions, but good application paper overall**

**Rating:** 8
**Confidence:** 4

**Review:**

General comment
==============
The authors used policy gradient optimization for generating RNA sequences that fold into a target secondary structure, reporting clear accuracy and runtime improvements over the previous state-of-the-art. The authors used BOHR for optimizing hyper-parameters and present a new dataset for evaluating RNA design methods. The paper is well motivated and mostly clearly written. However, the methodological contributions are limited and I have some important concerns about their evaluation. Overall, I feel it’s a good paper for an ICLR workshop or biological journal if the authors address the outstanding comments.

Major comments
=============
1. The methodological contributions are limited. The authors used existing approaches (policy gradient optimization and BOHR for hyperparameter optimization) but do not report new methods, e.g. for sequence modeling. Performing hyper-parameter optimization is in my eyes not novel, but common practice in the field. It would me more informative if the authors compared reinforcement learning to other approaches for (conditional) sequence generations, e.g. RNNs, autoregressive models, VAEs, or GANs, which have been previously reported for biological sequence generation (e.g. http://arxiv.org/abs/1804.01694).

2. Did the authors split all three datasets (Eterna, Rfam-Taneda, Rfam-learn-test) into train, eval, and test set, trained their method on the training set, optimized hyper-parameters on the eval set, and measured generalization and runtime on the test set? This is not described clearly enough in section 5. I suggest to summarize the number of sequences for each dataset and split in a table.

3. Did the authors also optimize the most important hyperparameters of RL-LS and other methods? Otherwise it is unclear if the performance gain is due to hyperparameter optimization or the method itself.

4. The time measurement (x-axis figure 3) is unclear. Is it the time that methods were given to solve a particular target structure and does figure 3 show the average number of solved structures in the test for a the time shown on the x-axis?

5. Were all methods compared on the same hardware (section 5; 20 cores; Broadwell E5-2630v4 2.2 GHz CPUs) and can they be parallelized over multiple CPU or GPU cores? This is essential for a fair runtime comparison.

6. The term ‘run’ (“unreliable outcomes in single runs”, section 4) is unclear. Is it a single sample from the model (one rollout), a particular hyperparameter configuration, or training the model once for a single target structure? This must be clarified for understanding the evaluation.

7. How does the accuracy and runtime or LEARNA scale depending on the sequence (structure) length?

8. How sensitive is the model performance depending on the context size k for representing the current state? Did the authors try to encode the entire target structure with, e.g. recurrent models, instead of using a window centered on the current position?

9. The authors should more clearly describe the local optimization step (section 3.1; reward). Were all nucleotides that differ mutated independently, or enumerated exhaustively? The latter would have a high runtime of O(3^d), where d is the number of nucleotides that differ. When do the authors start with the local optimization?

Minor comments
=============
10. The authors should replace ‘450x’ faster in the abstract by ‘clearly’ faster since the evaluation does not show that LEARNA is 450x faster than all other methods.

11. Does “At its most basic form” (introduction) mean that alternative RNA nucleotides exist? If so, this should be cited.

12. The authors should more clearly motive in the introduction why they created a new dataset.

13. The authors should mention in section 2.1 that the dot-bracket notation is not the only notation for representing RNA structures (https://www.tbi.univie.ac.at/RNA/ViennaRNA/doc/html/rna_structure_notations.html).

14. The authors should define the hamming distance (section 2.1). Do other distance metrics exist?

15. For the Traveling Salesman Problem (section 2.2) should the reward be the *negative* tour length?

16. The authors should more clearly describe the embedding layer (section 4). Are nucleotides one-hot encoded or represented as integers (0, 1  for ‘(‘ and ‘.’)?

---

> ### Author Response · Authors · 2018-11-23
> **Detailed replies 3/3**
>
> “8. How sensitive is the model performance depending on the context size \kappa for representing the current state? Did the authors try to encode the entire target structure with, e.g. recurrent models, instead of using a window centered on the current position?”
>
> --> Thanks for the suggestion. An RNN is already included in our search space, and was indeed selected by our joint architecture search and hyperparameter optimization. We have not yet experimented with encoding the entire target structure with an RNN, since having to backpropagate through that RNN at each time step of our agent would lead to a substantial increase of computational cost, be harder to train and increase the number of hyperparameters. Having said that, we do think this is a good idea if it can be made computationally efficient, e.g., by learning the embedding offline (although the training signal for that would need to be defined first); since this is not straightforward we leave it to future work.
>
> In terms of the importance of the context size, our new hyperparameter importance in Appendix I indicates that the context size (state space radius) \kappa does not appear to be very important.
>
>
> “9. The authors should more clearly describe the local optimization step (section 3.1; reward). Were all nucleotides that differ mutated independently, or enumerated exhaustively? The latter would have a high runtime of O(3^d), where d is the number of nucleotides that differ. When do the authors start with the local optimization?”
>
> --> We agree that the local improvement step should be described more clearly: we revised the reward paragraph and included pseudocode for computing the reward using the local improvement step (Appendix A). It works as follows: After the policy rollout we fold the candidate solution and compare it to the target structure, if less than \xi sites differ we perform this local improvement step in order to compute the reward. The value of \xi is not part of the hyperparameter optimization and based on the runtime costs and preliminary experiments we set xi=5, i.e., we used the local improvement step if the number of differing sites was at most 4. Keeping this number low was indeed important because of the computational complexity mentioned by the reviewer (it’s actually O(4^d), with d<=4).
>
>
> “10. The authors should replace ‘450x’ faster in the abstract by ‘clearly’ faster since the evaluation does not show that LEARNA is 450x faster than all other methods.”
>
> --> Thank you for the comment, we changed the abstract to say that our approach achieves new state-of-the-art performance on all benchmarks while also being orders of magnitudes faster in reaching the previous state-of-the-art performance. We note that these speedups (including the 450x one on the Eterna100 benchmark, Figure 3 (top)) can clearly be seen in the evaluation plots.
>
>
> “11. Does “At its most basic form” (introduction) mean that alternative RNA nucleotides exist? If so, this should be cited.”
>
> --> Thanks for this question. With “At its most basic form” we refer to the most basic structural form of RNA, which is a sequence of nucleotides. We have since clarified the phrasing to “At its most basic structural form”.
>
>
> “13. The authors should mention in section 2.1 that the dot-bracket notation is not the only notation for representing RNA structures (https://www.tbi.univie.ac.at/RNA/ViennaRNA/doc/html/rna_structure_notations.html).” and
> “14a. The authors should define the hamming distance (section 2.1).”
>
> --> We have included references, thank you for your comments.
>
>
> “14b. Do other distance metrics [than the hamming distance] exist?”
>
> --> While not formally metrics, we have experimented with the paired-unpaired-ratio and derivatives of the hamming distance. While also not a metric, the GC-content (which is the ratio of G and C nucleotides to the U and A nucleotides) has been used in the RNA Design literature (e.g. by antaRNA) as an additional objective.
>
>
> “15. For the Traveling Salesman Problem (section 2.2) should the reward be the *negative* tour length?”
>
> --> You are of course right, thank you for reading our paper carefully and bringing this to our attention; we fixed it.
>
>
> “16. The authors should more clearly describe the embedding layer (section 4). Are nucleotides one-hot encoded or represented as integers (0, 1 for ‘(‘ and ‘.’)?”
>
> --> Thank you for this comment; we agree and have included a clearer description. For representing nucleotides, our automated reinforcement learning approach includes the choice between: 1) a binary encoding differentiating between paired and unpaired sites, and 2) a learned embedding layer whose dimension is a hyperparameter (only active if the learned embedding is selected).
>
>
> Thanks again for all your comments! If we cleared up some of your concerns, we would kindly ask you to update your assessment.

---

> > ### Comment · AnonReviewer1 · 2018-11-27
> > **Manuscript clearly improved; only few minor comments.**
> >
> > I appreciate that you clearly addressed all comments and revised your manuscript! I have only few remaining comments.
> >
> > # 1a. Hyper-parameter optimization
> > I still believe that defining parameters of the neural network architectures in addition to optimization parameters is not a strong methodological contribution. This is rather common practice in reinforcement learning although often not described in detail in manuscripts. Methods for optimizing both discrete and continuous hyper-parameter had been described before, including Spearmint or Hyperopt. That said, I still believe that the pap1` er is a strong application paper!
> >
> > # 2. [Training/Validation/Test split of the data sets].
> > Do I understand you correctly that you proposed a ‘standard’ training, evaluation, and test set for Rfam-Learn, which does does not exist for Eterna100 or Rfam-Taneda? This is useful if the split is well defined (e.g. if the distribution of certain sequence properties is equal in all three sets), but not a strong contribution. Is the dataset larger than existing datasets, more diverse, or does it include additional sequences? I suggest to more clearly define differences in either the main text or appendix and more clearly motivate why Rfam-Learn is superior to existing datasets.
> >
> > # 3. Hyperparameter optimization
> > Please highlight in the main text that hyperparameters were only optimized for LEARNA and that other methods might also benefit by rigorously optimizing both model as well as optimization hyperparameters.
> >
> > # 7. How does the accuracy and runtime scale depending on the sequence (structure) length?
> > Thanks for the additional runtime analysis. Does each dot correspond to one target structure in the test set? Why are you showing the ‘minimum solution time’? Does the runtime vary over multiple runs? If so, it is more fair to show the average run time.
> >
> > # 8. How sensitive is the model performance depending on the context size \kappa for representing the current state?
> > Given that sequences can be hundreds of nucleotides long, I agree that RNNs would be slow and sensitive to exploding/vanishing gradients. You can consider non-recurrent models such as dilated CNNs or transformers in the future.
> >
> > Thanks clarifying that \kappa corresponds to the ‘state_radius’ in Appendix I. For consistency, I suggest to change the x-axis title to ‘state_radius \kappa’ or ‘\kappa’.
> >
> > # 9. Local improvement step
> > Thanks for clarifying the local improvement step (LIS). Figure 9 indicates LIS clearly boost performance, which is an important finding. Can you highlight this in the main text? Are other methods also likely to benefit from a post-hoc LIS?

---

> > > ### Author Response · Authors · 2018-11-29
> > > **Reply to minor comments 2/2**
> > >
> > > # 3. Hyperparameter optimization
> > > "Please highlight in the main text that hyperparameters were only optimized for LEARNA and that other methods might also benefit by rigorously optimizing both model as well as optimization hyperparameters.”
> > >
> > > ---> Thanks, we will definitely highlight in the final version which methods were optimized on what data set, and that other methods could benefit from that as well. (The server does not allow us to upload a new version at this time.)
> > >
> > >
> > > # 7. How does the accuracy and runtime scale depending on the sequence (structure) length?
> > > "Thanks for the additional runtime analysis. Does each dot correspond to one target structure in the test set? Why are you showing the ‘minimum solution time’? Does the runtime vary over multiple runs? If so, it is more fair to show the average run time.”
> > >
> > > ---> Yes, indeed, every point corresponds to a single target structure in the test set. We decided to plot the minimum (1) to account for missing data due to sequences not being solved within the time limit by individual runs and (2) since the minimum is used as the benchmarking criterion throughout the related RNA Design literature. However, we have addressed the average performance in Figure 3 (performance over time) which (also) shows the average solution time and in Tables 6-8 which list the number of solved sequences for various numbers of runs. Our approach performs well in both of these regards.
> > >
> > >
> > > # 8. How sensitive is the model performance depending on the context size \kappa for representing the current state?
> > > "Given that sequences can be hundreds of nucleotides long, I agree that RNNs would be slow and sensitive to exploding/vanishing gradients. You can consider non-recurrent models such as dilated CNNs or transformers in the future.”
> > >
> > > --->  Thanks, we will adjust the axis labels for the plots for the final version to be more consistent with the text. The sequence length is indeed challenging, and thank for your suggestion for future work, we’ll include these models in the ones we are planning to study next.
> > >
> > >
> > > # 9. Local improvement step
> > > "Thanks for clarifying the local improvement step (LIS). Figure 9 indicates LIS clearly boost performance, which is an important finding. Can you highlight this in the main text? Are other methods also likely to benefit from a post-hoc LIS?”
> > >
> > > ---> Thanks, we did mention the importance of this step in our ablation study in Section 6.2 where we discuss Figure 9, but will do so more explicitly in the final version. The way we view this step, it is a very limited local search applied when proposed sequences almost folds into the target structure; most of the other methods we compare against are local search methods that take a long time until they get close to the target structure; we would expect that applying this local improvement step would likely slow them down. We do, however, believe that this step could also benefit other generative models, such as MCTS-RNA, but we have not tried to incorporate it into MCTS-RNA; we will point out the possibility of doing so more explicitly in the final version.

---

> > > > ### Comment · AnonReviewer1 · 2018-12-01
> > > > **Increased rating**
> > > >
> > > > Thanks for your final answers and changes. I increased the rating of your paper to 8. Tl;DR;
> > > > * methodological contributions existing but incremental
> > > > * comprehensive evaluation and experiments
> > > > * strong application paper overall

---

> > > > > ### Comment · AnonReviewer1 · 2018-12-02
> > > > > **References hyperparameter optimization**
> > > > >
> > > > > References hyperparameter optimization:
> > > > > * https://arxiv.org/abs/1808.05377
> > > > > * https://arxiv.org/abs/1611.01578
> > > > > * https://arxiv.org/pdf/1807.07663.pdf

---

> > > > > > ### Author Response · Authors · 2018-12-07
> > > > > > **Thanks**
> > > > > >
> > > > > > Thanks for these references. We already cited the first two in Section 5
> > > > > > (Joint Architecture and Hyperparameter Search, top of page 6), but we now
> > > > > > think that a brief paragraph on architecture search and hyperparameter
> > > > > > optimization in the related work section would be useful as well, where all
> > > > > > these references will be a natural fit. Thanks again for the helpful
> > > > > > feedback and for increasing your rating!

---

> > > ### Author Response · Authors · 2018-11-29
> > > **Reply to minor comments 1/2**
> > >
> > > Thank you for appreciating our detailed rebuttal and our revised manuscript. We also thank you for again pointing out that our work is a strong application paper (as we mentioned in our top level comment, applications are specifically listed as relevant in the ICLR call for papers, including applications in computational biology and other fields).
> > >
> > >
> > > # 1a. Hyper-parameter optimization
> > > "I still believe that defining parameters of the neural network architectures in addition to optimization parameters is not a strong methodological contribution. This is rather common practice in reinforcement learning although often not described in detail in manuscripts. Methods for optimizing both discrete and continuous hyper-parameter had been described before, including Spearmint or Hyperopt. That said, I still believe that the paper is a strong application paper!”
> > >
> > > ---> We fully agree that hyperparameter optimization is an integral part of machine learning and reinforcement learning in general. For our application, it was the key to success, as we did not a priori know which architecture or state space size would work best. For this reason, we automatically searched a fairly flexible space that included pure RNNs, pure CNNs, and mixtures of these with an additional MLP. This level of parametrization is rarely laid out, so we do hope that you agree to at least some novelty in this regard. (Indeed, if you know of a reference that searched over a combination of RNNs and CNNs before we would be very grateful to know about it to not falsely claim novelty in this regard.)
> > >
> > > Spearmint and TPE are useful tools in general.  We expect that for our 14-dimensional space with many integer choices TPE would work better than Spearmint, and BOHB is a more efficient multi-fidelity variant of TPE (also see the BOHB paper for large speedups over TPE and Spearmint: proceedings.mlr.press/v80/falkner18a/falkner18a.pdf); our modest contribution in this regard is to provide a case study for this existing tool.
> > >
> > >
> > > # 2. [Training/Validation/Test split of the data sets].
> > > "Do I understand you correctly that you proposed a ‘standard’ training, evaluation, and test set for Rfam-Learn, which does does not exist for Eterna100 or Rfam-Taneda? This is useful if the split is well defined (e.g. if the distribution of certain sequence properties is equal in all three sets), but not a strong contribution. Is the dataset larger than existing datasets, more diverse, or does it include additional sequences? I suggest to more clearly define differences in either the main text or appendix and more clearly motivate why Rfam-Learn is superior to existing datasets.”
> > >
> > > ---> Yes, indeed, we proposed such a standard training/evaluation/test split, and these do not exist for Eterna100 or Rfam-Taneda. As described in the added Appendix C, we selected a subset of the Rfam database v13.0 based on difficulty (measured by number of known solutions and time it took MCTS-RNA to solve them) and controlled the distribution of sequence lengths across splits.
> > >
> > > Our data sets consist of 65000, 100, and 100 target structures (for training, validation, and test, respectively), based on naturally occurring RNA sequences. In contrast, Rfam-Taneda and Eterna100 contain only 29 and 100 sequences respectively. While the former also is a subset of the Rfam database, the latter consists of handcrafted sequences only. We included both in our work as they serve as the default test sets in the community. Our data sets are a “curated” selection of a larger corpus of natural RNA sequences allowing more data driven approaches to be applied. It is hard to compare RNA sequence datasets in terms of quantitative measures, but we tried to select an interesting collection that enables generalization across different RNA families. We hope this clarifies your questions regarding our new benchmark.

---

> ### Author Response · Authors · 2018-11-23
> **Detailed replies 2/3**
>
> “3. Hyperparameter optimization of other methods; Did the authors also optimize the most important hyperparameters of RL-LS and other methods? Otherwise it is unclear if the performance gain is due to hyperparameter optimization or the method itself.”
>
> --> We assess the performance of all methods on three test sets, where our method was trained and optimized using a single designated dataset for training and validation. The other methods we compare to either do not have clear/exposed hyperparameters (RNAinverse), were optimized by the original authors either also on a subset of the Rfam database (AntaRNA, and MCTS), or optimized on a non-disclosed dataset (RL-LS).
> Additionally, the authors of RL-LS, state in their paper: ”A more rigorous hyperparameter search might improve our results somewhat, but would probably not dramatically change the model's performance.”.
>
> Our empirical evaluation focuses more on generalization rather than optimizing the hyperparameters to every dataset. That is why we optimized each of our approaches (LEARNA and Meta-LEARNA) using only our own validation set. For our meta-learning approaches (Meta-LEARNA, Meta-LEARNA-Adapt) the single best configuration was then evaluated on the three test sets without modification and still surpassed the state-of-the-art. Potentially, all methods could be improved by further optimization on each type of dataset, but this was not our focus.
>
>
> “4. The time measurement (x-axis figure 3) is unclear. Is it the time that methods were given to solve a particular target structure and does figure 3 show the average number of solved structures in the test for the time shown on the x-axis?” and
> “6. The term ‘run’ (“unreliable outcomes in single runs”, section 4) is unclear. Is it a single sample from the model (one rollout), a particular hyperparameter configuration, or training the model once for a single target structure? This must be clarified for understanding the evaluation.”
>
> --> Thanks, you are right to point out that these two points were unclear. We believe this was due to an inconsistent usage of the term “run”. In Section 4 of our initial submission (joint architecture and hyperparameter optimization) we referred with “run” to a full optimization of the policy and in Section 5 of our initial submission (experiments) we referred with “run” to an “evaluation run” which consists of evaluating a given method once on each target structure in the corresponding benchmark. An evaluation run can be visualized by plotting the number of solved target structures across the time spent on each particular target structure. Existing benchmarks for RNA Design consider a number of evaluation runs and use the total number of target structures that were solved in at least one of these evaluation runs as the objective. Hence, Figure 3 visualizes aggregates of all evaluation runs: On the left side of Figure 3 we plot the total number of target structures that were solved in at least one evaluation run across time spent on each particular target structure, and similarly,  the right side of Figure 3 shows the average number of solved target structures. Thank you very much for pointing out this issue, we disambiguated the terms and worked on clarity.
>
>
> “5. Were all methods compared on the same hardware (section 5; 20 cores; Broadwell E5-2630v4 2.2 GHz CPUs) and can they be parallelized over multiple CPU or GPU cores? This is essential for a fair runtime comparison.”
>
> --> We agree that this is essential for a fair comparison and as we noted in the header of our experiments section in our initial submission all computations were done on the same listed CPU model. As mentioned in our initial submission, the training stage of Meta-LEARNA uses 20 cores (we use parallel PPO), but at validation/test time all methods were only allowed a single core (using core binding).
>
>
> “7. How does the accuracy and runtime scale depending on the sequence (structure) length?”
>
> --> Thank you for asking this important question. We have now included plots for solution times across sequence lengths (Appendix J), which clearly indicate that our approaches scale very well and are not affected a lot by increasing sequence length.

---

> ### Author Response · Authors · 2018-11-23
> **Detailed replies 1/3**
>
> Thanks for your positive feedback regarding our motivation and general writing, the characterization of our paper as a good application paper and for your comments, questions and helpful suggestions. In the following we reply to your comments and clarify some of the points:
>
>
> “1a. The methodological contributions are limited. Performing hyper-parameter optimization is in my eyes not novel, but common practice in the field.”
>
> --> We agree that hyperparameter optimization is clearly standard in RL, but our work goes much further than that. A joint optimization over neural architectures and hyperparameters, to the best of our knowledge, is novel in the field of RL (and is also not common in supervised learning). We would also like to repeat point (2) from our general reply to all reviewers concerning novelty, copied here for convenience:
>
> <“To the best of our knowledge, our paper is the first case study on the joint optimization of the architecture of the policy network (including both recurrent connections and convolutions in a single search space), the state representation, and the hyperparameters of an RL algorithm. In fact, we are not even aware of *any* other previous work on neural architecture search (NAS) for RL. Also, while there is of course a lot of work on NAS for CNNs and NAS for RNNs individually, we are not aware of any other previous NAS work that tackles a search space including both convolutions and recurrent units at the same time (i.e., with NAS choosing the best combination of the two). Finally, we are not aware of any previous work on NAS for meta-learning (other than learning a cell architecture and transferring that cell to a different dataset). We do believe that these are clear points in favor of our paper’s novelty, and we should have made these clearer in the submitted version of our paper; we’ve fixed this now in Section 5 and in the introduction.”>
>
>
> “1b. Related work;  It would me more informative if the authors compared reinforcement learning to other approaches for (conditional) sequence generations, e.g. RNNs, autoregressive models, VAEs, or GANs, which have been previously reported for biological sequence generation (e.g. http://arxiv.org/abs/1804.01694).”
>
> --> Thanks for the helpful comment on the interesting work in the fields of protein design and biological sequence generation. In our revised related work section we did include a discussion on the general field of matter engineering and reference a very recent review on generative approaches for this field. We did not experiment with VAEs or GANs (with appendix, our paper is already 30 pages...) but consider that future work. However, concerning RNNs, as described in Section 5, these were in fact part of our design space and were selected by the joint optimization process for two out of three final configurations used in our experiments (see Table 4 in Appendix A of our initial submission; in the revised version this is Table 5 in Appendix E).
>
>
> “2. [Training/Validation/Test split of the data sets]” and
> “12. The authors should more clearly motivate in the introduction why they created a new dataset.”
>
> --> The benchmarks used in the recent RNA Design literature Eterna100 (100 datapoints) and Rfam-Taneda (29 datapoints) do not have a train/validation/test split associated with them. (As ML researchers, we were surprised about this, too...) Hence, the need for a training and validation set of adequate size and diversity motivated us to introduce Rfam-Learn, which to the best of our knowledge is the first RNA Design benchmark with an explicit training/validation/test split.
>
> We optimized each of our approaches using only our own validation set (Rfam-Learn-Validation) and for our meta learning approach only used our own training set (Rfam-Learn-Train). To measure the final performance, as well as the transferability of the found architecture, hyperparameters, and the trained policy (Meta-LEARNA), the best configuration of each of our methods was then tested on Eterna100, Rfam-Taneda and Rfam-Learn-Test, and they achieve state-of-the-art results on all of them.
>
> We incorporated changes to clarify the above points and we thank you for the suggestion to use a table to display benchmark information as it indeed conveys the information more clearly.

---

### Official Review · AnonReviewer2 · 2018-11-05
**RNA sequence design with deep reinforcement learning**

**Rating:** 6
**Confidence:** 1

**Review:**

This work tackles the difficult RNA design problem, i.e. that of finding a RNA primary sequence that is going to fold into a secondary/tertiary structure able to perform a desired biological function. More specifically, it used Reinforcement Learning (RL) to find the best sequence that will fold into a target secondary structure, using the Zuker algorithm and designing a new primary sequence 'from scratch'. A new benchmark data set is also introduced in the paper along .

Questions/remarks:
 - I struggle with your notations as soon as section 2.1. What is the star (*) superscript for? Was expecting the length of the RNA sequence instead. Same on p4, when introducing the notation of your decision process $ D_w $, explicitly introduce all the ingredients.
 - in Equation (2) on p4, maybe clarify the notation with '.', '(' and ')' for example as the reader could really struggle.
 - I didn't really understand the message in Section 4, not being an expert in the field. Could you clarify your contribution here?
 - your 'Ablation study' in Section 5.2; does it correspond to true uncertainty/noise that could be observed in real data?
 - why a new benchmark data set, when there exist good ones to compare your method to, e.g. in competitions like CASP for proteins?
 - do you make your implementation available?
 - quite like the clarification of the relationship of your work to that of Eastman et al. 2018. Could you also include discussions to other papers, e.g. Chuai et al. 2018 Genome Biol and Shi et al. 2018 SentRNA on arXiv?

Altogether the paper reads well, seems to have adequate references, motivates and proposes 3 variations of a new algorithm for a difficult learning problem. Not being an expert in the field, I just can't judge about the novelty of the appraoch.

---

> ### Author Response · Authors · 2018-11-23
> **Detailed replies**
>
> Thanks for the suggested improvements, the insightful comments and questions! Thanks also for the positive feedback on the text of the paper, references and motivation. In the following we provide detailed replies:
>
>
> “1.  What is the star (*) superscript for? Was expecting the length of the RNA sequence instead.”
>
> --> Thank you for pointing out this undefined and potentially confusing use of notation. The Kleene Operator (*) applied to a set M yields a set of all finite-length sequences based on M, and we used it since RNA Structures have variable length. But we do agree that this can be confusing and made changes to talk about a specific structure w and then use N^|w| as you suggested.
>
>
> “2. Same on p4, when introducing the notation of your decision process $ D_w $, explicitly introduce all the ingredients.”
>
> --> We agree with you and revised the definition of the undiscounted decision process. We now explicitly name the components of the quadruple D_w and also refer to the specifics in the paragraphs following the definition of D_w.
>
>
> “3. in Equation (2) on p4, maybe clarify the notation with '.', '(' and ')' for example as the reader could really struggle.”
>
> --> We have looked at this again and changed the equation, making it easier to parse for the reader. We have also included a verbatim “dot” and “opening bracket” to not confuse the reader by the notation.
>
>
> “4. I didn't really understand the message in Section 4, not being an expert in the field. Could you clarify your contribution here?”
>
> --> Thanks for asking about this! As detailed in our general reply to all reviewers, this section breaks novel ground concerning the joint optimization of neural architectures and hyperparameters, joint search over combinations of recurrent and convolutional layers in the same search space, neural architecture search for RL, and neural architecture search for meta-learning. In the interest of brevity, we refer to the detailed reply to all reviewers above.
>
>
> “5. your 'Ablation study' in Section 5.2; does it correspond to true uncertainty/noise that could be observed in real data?”
>
> --> In our ablation study, we disable one functional component of our approach at a time in order to study its influence; incorporating ablations in empirically evaluated work is important to find out whether all proposed components are necessary and contribute to the final performance. Our ablation study is performed on the test split of our introduced dataset, which as we point out in the heading of Section 5 of our initial submission, has been generated from sequences observed in living organisms as listed in the Rfam 13.0 database; it is not used to optimize hyperparameters but is a post hoc evaluation.
>
>
> “6. why a new benchmark data set, when there exist good ones to compare your method to, e.g. in competitions like CASP for proteins?”
>
> --> We report our results on two widely used benchmarks which were also used in the work we compare to but unfortunately only provide test sets (no training/validation/test split). To the best of our knowledge, we introduce the first benchmark with an explicit training/validation/test split. The reviewer is right in that there exist other and good data sources, but to the best of our knowledge not in the form of competitions. To mention two databases by name:
>
> * the STRAND database (http://www.rnasoft.ca/strand/) that currently holds 4666 known RNA secondary structures
> * the FRABASE 2.0 database (https://bmcbioinformatics.biomedcentral.com/articles/10.1186/1471-2105-11-231) with 2753 entries of fragments of secondary structures
>
> Both databases have not been used by the publications we compare to and cannot satisfy the size and sequence diversity requirements for our meta-learning approach and future research (especially for methods needing a large training set). The Rfam 13.0 database we use here for generating our new training-, validation- and test set is large enough to yield three distinct datasets of meaningful sizes and diversity.
>
>
> “7. do you make your implementation available?”
>
> --> Thanks for the question, indeed, we strongly believe in sharing code (as well as data) to reproduce scientific findings. To stand by this opinion, we had included a note in the conclusion of our initial submission that we will make all of our code and data available upon acceptance of our paper.
>
>
> “8. quite like the clarification of the relationship of your work to that of Eastman et al. 2018. Could you also include discussions to other papers, e.g. Chuai et al. 2018 Genome Biol and Shi et al. 2018 SentRNA on arXiv”
>
> --> Thanks for the positive feedback regarding our discussion of the relationship of our work to that of Eastman et al. 2018, and for bringing the related work to our attention. We included discussions in our related work section.
>
>
> Thanks again for all your comments! If we cleared up some of your concerns, we would kindly ask you to update your assessment.

---

### Official Review · AnonReviewer3 · 2018-11-07
**Interesting application of RL to DNA, new SotA perf, some theoretical novelty**

**Rating:** 6
**Confidence:** 4

**Review:**

I'm happy with the revisions the authors have made, as I find that they call out the novel contributions a bit more explicitly. Specifically I see some novel work in the area of simultaneous multi-task/meta-RL and black box optimization of the policy net architectures. I don't think calling this NAS is justified; calling it bayesopt or black box opt is fair. NAS uses a neural net to propose experiments over structured graphs of computation nodes. This work appears to be simpler hyperparameter optimization.

====

Quality:
The work is well done, and the experiments are reasonable/competitive, showcasing other recent work and outperforming.

Clarity:
I thought the presentation was tolerable. I was a bit confused by Table 1 until reading the prose at the bottom of page 7 indicated Table 1 is presenting percentages, not integer quantities. The local improvement step is not very clearly explained. Are all combos tried across all mismatched positions, or do we try each mismatched position independently holding the others to their predicted values? What value of zeta did you end up using? It seems like this is essential to getting good performance. It is completely unclear to me what the 'restart option' does.

Originality:
Using RL in this specific application setting seems relatively new (though also explored by RL-LS in https://www.ncbi.nlm.nih.gov/pmc/articles/PMC6029810/). On the other hand, the approach used doesn't seem to be substantially different than anything else typically used for policy gradient RL. The meta-learning approach is interesting, though again not too different from multi-task approaches (though these are perhaps less common in RL than in general deep learning).

Significance:
Likely to be of practical utility in the inverse design space, specifically therapeutics, CRISPR guide RNA design, etc. Interesting to ICLR as an application area but probably not much theory/methods interest.


On balance I lean slightly against accepting and think this is a better fit to either a workshop or a more domain-specific venue (MLHC http://mucmd.org/ for example).

---

> ### Author Response · Authors · 2018-11-23
> **Detailed replies**
>
> Thanks for your helpful comments and questions. Thanks also for your positive feedback on our work in general, our experiments, the significance of our approach for therapeutics and other practical use cases and for characterizing our work as interesting to ICLR as an application area. We would like to comment on your suggestions, comments and questions in the following.
>
>
> “1. I was a bit confused by Table 1 until reading the prose at the bottom of page 7 indicated Table 1 is presenting percentages, not integer quantities.”
>
> --> Reviewing Table 1, we agree that it could be confusing -- its caption did mention that all entries represent percentages and not total values, but this was unnecessarily indirect, and we now reworked the tables to include a percentage symbol to make it clearer.
>
>
> “2. The local improvement step is not very clearly explained. Are all combos tried across all mismatched positions, or do we try each mismatched position independently holding the others to their predicted values? What value of \xi did you end up using? It seems like this is essential to getting good performance.”
>
> --> Thanks, we agree that the local improvement step should be described more clearly and that it is an important part of our approach (as the empirical evidence in our ablation study suggests). We have since reworked the corresponding paragraph and included pseudocode (Appendix A). It works as follows: we exhaustively try all possible nucleotide assignments for the mismatched positions which takes at most 4^|differing_sites| additional folds. The value of \xi we used was 5, i.e., we used the local improvement step if the number of differing sites was at most 4. This was set early on based on runtime considerations and preliminary experiments and was not part of our hyperparameter optimization; thank you for the detailed reading of our paper and pointing out this missing value, we have added it now.
>
>
> “3. It is completely unclear to me what the 'restart option' does.”
>
> --> Thanks for pointing out this missing information. Since RL algorithms are prone to getting stuck in local minima, we decided to employ occasional restarts (i.e., reinitialization) in our strategies. We now describe this in the revised version in Section 5. For LEARNA and for Meta-LEARNA-Adapt, this makes a difference, whereas for Meta-LEARNA it does not since Meta-LEARNA is directly sampling from the model without updating it (which is equivalent to restarting at each step)
>
>
> “4. Using RL in this specific application setting seems relatively new (though also explored by RL-LS in https://www.ncbi.nlm.nih.gov/pmc/articles/PMC6029810/).”
>
> --> Thanks for this comment! Indeed, the reinforcement learning guided local search (RL-LS) was developed in parallel and independently from LEARNA (as mentioned in our discussion on RL-LS in Section 2.2 of our initial submission; now discussed in Section 3). However, the two approaches differ a lot: although both approaches employ RL to RNA Design, Eastman et al. follows the common approach of using a local search strategy for solving the RNA Design problem, while we try to tackle the problem with a generative model.
>
>
> “5. On the other hand, the approach used doesn't seem to be substantially different than anything else typically used for policy gradient RL. The meta-learning approach is interesting, though again not too different from multi-task approaches (though these are perhaps less common in RL than in general deep learning).”
>
> --> We agree that the policy gradient approach we use is standard, but that using meta-learning in this context is already less common. We would also like to repeat the point concerning novelty of our joint optimization we made in the general reply to all reviewers. We copied this here for convenience:
>
> <“To the best of our knowledge, our paper is the first case study on the joint optimization of the architecture of the policy network (including both recurrent connections and convolutions in a single search space), the state representation, and the hyperparameters of an RL algorithm. In fact, we are not even aware of *any* other previous work on neural architecture search (NAS) for RL. Also, while there is of course a lot of work on NAS for CNNs and NAS for RNNs individually, we are not aware of any other previous NAS work that tackles a search space including both convolutions and recurrent units at the same time (i.e., with NAS choosing the best combination of the two). Finally, we are not aware of any previous work on NAS for meta-learning (other than learning a cell architecture and transferring that cell to a different dataset). We do believe that these are clear points in favor of our paper’s novelty, and we should have made these clearer in the submitted version of our paper; we’ve fixed this now in Section 5 and in the introduction.”>
>
>
> Thanks again for your comments! If we cleared up some of your concerns, we would kindly ask you to update your assessment.

---

> ### Author Response · Authors · 2018-11-29
> **Reply to update**
>
> “I'm happy with the revisions the authors have made, as I find that they call out the novel contributions a bit more explicitly. Specifically I see some novel work in the area of simultaneous multi-task/meta-RL and black box optimization of the policy net architectures. I don't think calling this NAS is justified; calling it bayesopt or black box opt is fair. NAS uses a neural net to propose experiments over structured graphs of computation nodes. This work appears to be simpler hyperparameter optimization.”
>
> --> Thanks for the positive feedback, and for seeing some novel work in the area of simultaneous multi-task/meta-RL and black box optimization of the policy net architectures. We agree that much of the current work on NAS does indeed use neural nets to propose experiments over structured graphs of computation nodes, and to not be confusing we’ll reword. For completeness, we would like to mention, however, that not all NAS methods fall into that category; specifically, most current NAS papers use a cell search space of fixed dimensionality, and the method that has the best published performance (regularized evolution, by Quoc Le’s group at Google Brain [https://arxiv.org/abs/1802.01548], better than reinforcement learning by the same group and others) does *not* use a neural network but a simpler hyperparameter optimization method with a fixed dimensionality approach through genetic algorithms. But this is really not important for this paper and we will simply reword to the non-contentious term “joint optimization of architectural choices, state description hyperparameters, and RL algorithm hyperparameters”.
>
> Thanks again for the positive reply and update!

---

> > ### Comment · AnonReviewer3 · 2018-12-04
> > **Parentheses**
> >
> > Thanks for updating the text to avoid confusion.
> >
> > I suppose the interpretation is dependent on parentheses. :)
> > (Neural Architecture) Search is [presumably] yours
> > Neural (Architecture Search) is mine
> >
> > I would argue that the paper by Quoc Le's group does not call their method NAS. Rather they frame it as 'searching the NASNet space' with an evolutionary strategy.

---

> > > ### Author Response · Authors · 2018-12-07
> > > **Indeed, parentheses :)**
> > >
> > > Thanks, we did not think of the interpretation "Neural (Architecture Search)". Now being very aware of the two different possible interpretations of NAS, we will be sure to use a wording that avoids the confusion. Thanks again!

---

### Author Response · Authors · 2018-11-23
**Aspect of being an application paper and novelty of our methods**

We would like to thank all reviewers for their helpful comments! In response to them we performed additional analysis, updated the paper and now reply to all reviews at the same time in order to limit the overhead for the reviewers.

Since these were comments several reviewers had, we would like to comment on (1) the aspect of being an application paper and (2) the novelty of our methods.

(1) We are glad that several reviewers found the application we are tackling interesting. We would like to note that applications are specifically listed as relevant in the ICLR call for papers (https://iclr.cc/Conferences/2019/CallForPapers), including applications in computational biology and others. We believe that a strong application paper takes existing methods and applies them to an interesting and difficult problem of a certain significance. In the process, the formulation of the problem, and technical details need to be adjusted to make it work. Additionally, a thorough evaluation comparing the method to other state-of-the-art approaches from the field and analyzing the importance of components (e.g. via ablation) is vital. We feel that we accomplished these in our work, and our reviews also indicate that the reviewers agree.

(2) Having said that, we in fact also believe that our work is novel in many ways other than this application. While hyperparameter optimization is clearly standard in RL, to the best of our knowledge, our paper is the first case study on the joint optimization of the architecture of the policy network, the state representation, and the hyperparameters of an RL algorithm. In fact, we are not even aware of *any* other previous work on neural architecture search (NAS) for RL. Also, while there is of course a lot of work on NAS for CNNs and NAS for RNNs individually, we are not aware of any other previous NAS work that tackles a search space including both convolutions and recurrent units at the same time (i.e., with NAS choosing the best combination of the two). Finally, we are not aware of any previous work on NAS for meta-learning (other than learning a cell architecture and transferring that cell to a different dataset). We do believe that these are clear points in favor of our paper’s novelty, and we agree with the reviewers that we should have made these much clearer in the submitted version of our paper; we are thankful to the reviewers’ comments and have fixed this now.

We would also like to note that, e.g., the popular population-based-training (PBT) method for tuning RL hyperparameters, is limited to optimizing hyperparameters that can be adapted during the optimization trajectory, while our approach also handles the tuning of other choices, such as the neural architecture and the state representation. As such, our paper can be viewed as an important step towards “automated reinforcement learning”, applied to a real-world problem (which we also believe to be novel).


We made the following changes to the paper in response to the reviews:

* Relating to (2) above, we clarified the novelty of our joint and automated architecture and hyperparameter search and added subsections to distinguish between the search space and the search procedure in the corresponding section.

* We added a parameter importance analysis to Section 6 (experiments) which supports the importance of the joint optimization of the policy network’s architecture, the environment parameters and the training hyperparameters.

* We explained our experimental protocol better, including more details on the used datasets from the literature and the dataset we compiled ourselves.

* We split our previous background section into two distinct sections, one for explaining the RNA-Design problem and one for discussing related work.

* We restructured the appendix, included plots that compare the performance of all approaches across different sequence lengths (Appendix J) and show the strong scaling of our approaches with sequence length, and added more analysis regarding our joint architecture and hyperparameter optimization.

* We incorporated clarification and discussion where indicated by the reviewers. We detail these changes in our responses to the individual reviewers.

---

### Meta-Review · Area_Chair1 · 2018-12-15
**Consensus is accept**

**Confidence:** 5
**Recommendation:** Accept (Poster)

**Metareview:**

After a healthy discussion between reviewers and authors, the reviewers' consensus is to recommend acceptance to ICLR. The authors thoroughly addressed reviewer concerns, and all reviewers noted the quality of the paper, methodological innovations and SotA results.